# Localized Meta-Learning: A PAC-Bayes Analysis for Meta-Learning Beyond Global Prior

## Abstract

Meta-learning methods learn the meta-knowledge among various training tasks and aim to promote the learning of new tasks under the task similarity assumption. However, such meta-knowledge is often represented as a fixed distribution, which is too restrictive to capture various specific task information. In this work, we present a localized meta-learning framework based on the PAC-Bayes theory. In particular, we propose an LCC-based prior predictor that allows the meta learner to adaptively generate local meta-knowledge for specific tasks. We further develop a practical algorithm with deep neural network based on the bound. Empirical results on real-world datasets demonstrate the efficacy of the proposed method.

## 1 Introduction

Recent years have seen a resurgence of interest in the field of meta-learning, or *learning-to-learn* (Thrun & Pratt, 2012), especially for empowering deep neural networks the capability of fast adapting to unseen tasks just as humans (Finn et al., 2017; Ravi & Larochelle, 2017). More concretely, the neural networks are trained from a sequence of datasets, associated with different learning tasks sampled from a meta-distribution (also called task environment (Baxter, 2000; Maurer, 2005)). The principal aim of meta learner is to extract transferable meta-knowledge from observed tasks and facilitate the learning of new tasks sampled from the same meta-distribution. The performance is measured by the generalization ability from a finite set of observed tasks, which is evaluated by learning related unseen tasks. For this reason, there has been considerable interest in theoretical bounds on the generalization in terms of the meta-learning algorithm (Denevi et al., 2018b;a).

One typical line of work (Pentina & Lampert, 2014; Amit & Meir, 2018) use PAC-Bayes bound to analyze the generalization behavior of the meta learner and quantify the relation between the expected loss on new tasks and the average loss on the observed tasks. In this setup, we formulate meta-learning as hierarchical Bayes. Accordingly, meta-knowledge is instantiated as a global distribution over all possible priors, which we call *hyperprior* and is chosen before observing training tasks. Each *prior* is a distribution over a family of classifiers w.r.t. a particular task. To learn versatile meta-knowledge across tasks, the meta learner observes a sequence of training tasks and adjusts its hyperprior into a *hyperposterior* distribution over the set of priors. To solve a new task, the base learner produces a *posterior* distribution over a family of classifiers based on the associated sample set and the prior generated by the hyperposterior.

However, such meta-knowledge is shared across tasks. The global hyperposterior is rather generic, typically not well-tailored to various specific tasks. Consequently, it leads to sub-optimal performance for any individual prediction task. As a motivational example, suppose we have two different tasks: distinguishing motorcycle versus bicycle and distinguishing motorcycle versus car. Intuitively, each task uses distinct discriminative patterns and thus the desired meta-knowledge is required to extract these patterns simultaneously. This could be a challenging problem to represent it with a global hyperposterior since the most significant patterns in the first task could be irrelevant or even detrimental to the second task.

Hence, we are motivated to pursue a meta-learning framework to effectively define the hyperposterior. The inspiration comes from the PAC-Bayes literature on data distribution dependent priors (Catoni, 2007; Parrado-Hernández et al., 2012; Dziugaite & Roy, 2018). The choice of posterior in each task is constrained by the need to minimize the relative entropy between prior and posterior since this divergence forms part of the bound and is typically large in standard PAC-Bayes

44 approaches (Lever et al., 2013). Thus, choosing an appropriate prior for each task which is close to
45 the related posterior could yield improved generalization bounds.

46 Inspired by this, we propose a **Localized Meta-Learning** (LML) framework. Instead of formulating
47 meta-knowledge as a global hyperposterior, we learn a conditional hyperposterior given task data
48 distribution that allows a meta learner to adaptively generate an appropriate prior for a new task.
49 However, the task data distribution is unknown, and our only perception for it is via the associated
50 sample set. Nevertheless, if the conditional hyperposterior is relatively stable to perturbations of the
51 sample set, then the generated prior could still reflect the underlying task data distribution, resulting
52 in a generalization bound that still holds with smaller probability. Following this intuition, the
53 dependence of a conditional hyperposterior on the task data distribution is parameterized by a prior
54 predictor using Local Coordinate Coding (LCC)(Yu et al., 2009). In particular, if the classifier in
55 each task is specialized to a parametric model, including deep neural network, the proposed LCC-
56 based prior predictor predicts the model parameters using the sample set by exploiting the local
57 information on the latent manifold. LCC-based prior predictor is invariant under permutations of its
58 inputs and could be further used for unseen tasks.

59 The main contributions of this work include: (i) We present a localized meta-learning framework
60 which provides a means to tighten the original PAC-Bayes meta-learning bound (Pentina & Lam-
61 pert, 2014; Amit & Meir, 2018) by minimizing the task-complexity term by choosing data-dependent
62 prior; (ii) We propose an LCC-based prior predictor, an implementation of conditional hyperposte-
63 rior, to generate local meta-knowledge for specific task; (iii) We derive a practical localized meta-
64 learning algorithm for deep neural networks by minimizing the bound; (iv) Experimental results
65 demonstrate improved performance over meta-learning method in this field.

## 2 PRELIMINARIES

### 2.1 LOCAL COORDINATE CODING

68 We first review some definitions of Local Coordinate Coding (LCC) (Yu et al., 2009) based on which
69 we develop the proposed LCC-based prior predictor.

70 **Definition 1.** *(Lipschitz Smoothness (Yu et al., 2009).) A function $f(\mathbf{x})$ on $\mathbb{R}^d$ is a $(\alpha, \beta)$-Lipschitz*
71 *smooth w.r.t. a norm $\|\cdot\|$ if $\|f(\mathbf{x}) - f(\mathbf{x}')\| \le \alpha \|\mathbf{x} - \mathbf{x}'\|$ and $\|f(\mathbf{x}') - f(\mathbf{x}) - \nabla f(\mathbf{x})^\top (\mathbf{x}' - \mathbf{x})\| \le$*
72 $\beta \|\mathbf{x} - \mathbf{x}'\|^2$.

73 **Definition 2.** *(Coordinate Coding (Yu et al., 2009).) A coordinate coding is a pair $(\boldsymbol{\gamma}, C)$, where*
74 *$C \subset \mathbb{R}^d$ is a set of anchor points, and $\boldsymbol{\gamma}$ is a map of $\mathbf{x} \in \mathbb{R}^d$ to $[\gamma_{\mathbf{u}}(\mathbf{x})]_{\mathbf{u} \in C} \in \mathbb{R}^{|C|}$ such*
75 *that $\sum_{\mathbf{u}} \gamma_{\mathbf{u}}(\mathbf{x}) = 1$. It induces the following physical approximation of $\mathbf{x}$ in $\mathbb{R}^d$ : $\gamma(\mathbf{x}) =$*
76 $\sum_{\mathbf{u} \in C} \gamma_{\mathbf{u}}(\mathbf{x})\mathbf{u}$.

**Definition 3.** *(Latent Manifold (Yu et al., 2009).) A subset $\mathcal{M} \subset \mathbb{R}^d$ is called a smooth manifold*
*with an **intrinsic dimension** $d := d_{\mathcal{M}}$ if there exists a constant $c_{\mathcal{M}}$ such that given any $\mathbf{x} \in \mathcal{M}$,*
*there exists $d$ bases $\mathbf{u}_1(\mathbf{x}), \ldots, \mathbf{u}_d(\mathbf{x}) \in \mathbb{R}^d$ so that $\forall \mathbf{x}' \in \mathcal{M}$:*

$$\inf_{\boldsymbol{\gamma} \in \mathbb{R}^{|C|}} \|\mathbf{x}' - \mathbf{x} - \sum_{j=1}^{d} \gamma_j \mathbf{u}_j(\mathbf{x})\|_2 \le c_{\mathcal{M}} \|\mathbf{x}' - \mathbf{x}\|_2^2,$$

77 *where $\boldsymbol{\gamma} = [\gamma_1, \ldots, \gamma_d]^\top$ are the local codings w.r.t. the bases.*

**Definition 4.** *(Covering Number (Yu et al., 2009).) The covering number $\mathcal{N}(\epsilon, \mathcal{M})$ is the smallest*
*cardinality of an $\epsilon$-ccover $C \subset \mathcal{M}$. That is,*

$$\sup_{\mathbf{x} \in \mathcal{M}} \inf_{\mathbf{v} \in C} \|\mathbf{x} - \mathbf{v}\| \le \epsilon.$$

78 Definition 2 and 3 imply that any point in $\mathbb{R}^d$ can be expressed as a linear combination of a set of
79 anchor points. Later, we will use them to develop the prior predictor.

### 2.2 PAC-BAYES META-LEARNING

81 In order to present the advances proposed in this paper, we next recall some definitions in PAC-Bayes
82 theory for single-task learning and meta-learning (Catoni, 2007; Baxter, 2000; Pentina & Lampert,

2014; Amit & Meir, 2018). In the context of classification, we assume all tasks share the same input space $\mathcal{X}$, output space $\mathcal{Y}$, space of classifiers (hypotheses) $\mathcal{H} \subset \{h : \mathcal{X} \to \mathcal{Y}\}$ and loss function $\ell : \mathcal{Y} \times \mathcal{Y} \to [0, 1]$. The meta learner observes $n$ tasks in the form of sample sets $S_1, \ldots, S_n$. The number of samples in task $i$ is denoted by $m_i$. Each observed task $i$ consists of a set of i.i.d. samples $S_i = \{(\mathbf{x}_j, y_j)\}_{j=1}^{m_i}$, which is drawn from a data distribution $S_i \sim D_i^{m_i}$. Following the meta-learning setup in (Baxter, 2000), we assume that each data distribution $D_i$ is generated i.i.d. from the same meta distribution $\tau$. Let $h(\mathbf{x})$ be the prediction of $\mathbf{x}$, the goal of each task is to find a classifier $h$ that minimizes the expected loss $\mathbb{E}_{\mathbf{x} \sim D} \ell(h(\mathbf{x}), y)$. Since the underlying 'true' data distribution $D_i$ is unknown, the base learner receives a finite set of samples $S_i$ and produces an "optimal" classifier $h = A_b(S_i)$ with a deterministic learning algorithm $A_b(\cdot)$ that will be used to predict the labels of unseen inputs.

PAC-Bayes theory studies the properties of randomized classifier, called Gibbs classifier. Let $Q$ be a posterior distribution over $\mathcal{H}$, to make a prediction, the Gibbs classifier samples a classifier $h \in \mathcal{H}$ according to $Q$ and then predicts a label with the chosen $h$. The expected error under data distribution $D$ and empirical error on the sample set $S$ are then given by averaging over distribution $Q$, namely $er(Q) = \mathbb{E}_{h \sim Q} \mathbb{E}_{(x,y) \sim D} \ell(h(x), y)$ and $\widehat{er}(Q) = \mathbb{E}_{h \sim Q} \frac{1}{m} \sum_{j=1}^{m} \ell(h(x_j), y_j)$, respectively. Then, we can get the following PAC-Bayes generalization bound of Catoni (2007) in a simplified form suggested by Germain et al. (2009).

**Theorem 1.** *(Catoni's bound) Let $P$ be some prior distribution over $\mathcal{H}$. Then for any $\delta \in (0, 1]$, and any real number $c > 0$, the following inequality holds uniformly for all posteriors distribution $Q$ with probability at least $1 - \delta$,*

$$er(Q) \le \frac{c}{1 - e^{-c}} \left[ \widehat{er}(Q) + \frac{KL(Q||P) + \log \frac{1}{\delta}}{mc} \right]. \tag{1}$$

The PAC-Bayes bound holds uniformly for all $Q$, it also holds for the data dependent $Q$. By choosing the posterior $Q$ that minimizes the PAC-Bayes bound, we obtain an learning algorithm with generalization guarantees. Note that the value $c$ allows to control the trade-off between the empirical error and the complexity term.

The goal of the meta learner is to extract meta-knowledge contained in the observed tasks that will be used as prior knowledge for learning new tasks. The prior knowledge $P$ is in the form of a distribution over classifiers $\mathcal{H}$. In each task, the base learner produces a posterior $Q = A_b(S, P)$ over $\mathcal{H}$ based on a sample set $S$ and a prior $P$. All tasks are learned through the same learning procedure. The meta learner treats the prior $P$ itself as a random variable and assumes the meta-knowledge is in the form of a distribution over all possible priors. Let hyperprior $\mathcal{P}$ be an initial distribution over priors, meta learner uses the observed tasks to adjust its original hyperprior $\mathcal{P}$ into hyperposterior $\mathcal{Q}$ from the learning process. The quality of the hyperposterior $\mathcal{Q}$ is measured by the expected task error of learning new tasks using priors generated from it, which is formulated as:

$$er(\mathcal{Q}) = \mathbb{E}_{P \sim \mathcal{Q}} \mathbb{E}_{(D,m) \sim \tau, S \sim D^m} er(Q = A_b(S, P)). \tag{2}$$

Accordingly, the empirical counterpart of the above quantity is given by:

$$\hat{er}(\mathcal{Q}) = \mathbb{E}_{P \sim \mathcal{Q}} \frac{1}{n} \sum_{i=1}^{n} \hat{er}(Q = A_b(S_i, P)). \tag{3}$$

## 3 PAC-BAYES META-LEARNING BOUND WITH GAUSSIAN RANDOMIZATION

Based on the above definition, Pentina & Lampert (2014) and Amit & Meir (2018) present meta-learning PAC-Bayes generalization bounds w.r.t. hyperposterior $\mathcal{Q}$ by using McAllester's single-task bound (McAllester, 1999). Here we present a new meta-learning generalization bound with Gaussian randomization by using Catoni's bound in Eq. (1). In particular, the classifier $h$ is parameterized as $h_{\mathbf{w}}$ with $\mathbf{w} \in \mathbb{R}^{d_{\mathbf{w}}}$. The prior and posterior is a distribution over the set of all possible parameters $\mathbf{w}$. We choose both the prior $P$ and posterior $Q$ to be spherical Gaussians, i.e. $P = \mathcal{N}(\mathbf{w}^P, \sigma_{\mathbf{w}}^2 I_{d_{\mathbf{w}}})$ and $Q = \mathcal{N}(\mathbf{w}^Q, \sigma_{\mathbf{w}}^2 I_{d_{\mathbf{w}}})$. The mean $\mathbf{w}^P$ is a random variable distributed first according to the hyperprior $\mathcal{P}$, which we formulate as $\mathcal{N}(0, \sigma_{\mathbf{w}}^2 I_{d_{\mathbf{w}}})$, and later according to hyperposterior $\mathcal{Q}$, which we model as $\mathcal{N}(\mathbf{w}^{\mathcal{Q}}, \sigma_{\mathbf{w}}^2 I_{d_{\mathbf{w}}})$. When encountering a new task $i$, we first sample the mean of prior

$\mathbf{w}_i^P$ from the hyperposterior $\mathcal{N}(\mathbf{w}^{\mathcal{Q}}, \sigma_{\mathbf{w}}^2 I_{d_{\mathbf{w}}})$, and then use it as a basis to learn the mean of posterior $\mathbf{w}_i^Q = A_b(S_i, P)$, as shown in Figure 1(left). Then, we could derive the following PAC-Bayes meta-learning bound.

**Theorem 2.** *Consider the Meta-Learning (ML) framework, given the hyperprior $\mathcal{P} = \mathcal{N}(0, \sigma_{\mathbf{v}}^2 I_{d_{\mathbf{v}}})$, then for any hyperposterior $\mathcal{Q}$, any $c_1, c_2 > 0$ and any $\delta \in (0, 1]$ with probability $\geq 1 - \delta$ we have,*

$$
\begin{aligned}
er(\mathcal{Q}) \leq & c_1' c_2' \hat{er}(\mathcal{Q}) + (\sum_{i=1}^{n} \frac{c_1' c_2'}{2c_2 n m_i \sigma_{\mathbf{w}}^2} + \frac{c_1'}{2c_1 n \sigma_{\mathbf{w}}^2}) \|\mathbf{w}^{\mathcal{Q}}\|^2 + \sum_{i=1}^{n} \frac{c_1' c_2'}{2c_2 n m_i \sigma_{\mathbf{w}}^2} \| \mathbb{E}_{\mathbf{w}^P} \mathbf{w}_i^Q - \mathbf{w}^{\mathcal{Q}} \|^2 \\
& + \sum_{i=1}^{n} \frac{c_1' c_2'}{c_2 n m_i \sigma_{\mathbf{w}}^2} (\frac{1}{2} + \log \frac{2n}{\delta}) + \frac{c_1'}{c_1 n \sigma_{\mathbf{w}}^2} \log \frac{2}{\delta},
\end{aligned}
\tag{4}
$$

*where $c_1' = \frac{c_1}{1 - e^{-c_1}}$ and $c_2' = \frac{c_2}{1 - e^{-c_2}}$.*

*Proof.* See Appendix B.3 for the proof.

Notice that the expected task generalization error is bounded by the empirical multi-task error plus two complexity terms. The first term demonstrates the environment-complexity which converges to zero if an infinite number of tasks are observed from the task environment ($n \to \infty$), while the second is the task-complexity of the observed tasks which converges to zero when the sufficient samples in each task is observed ($m_i \to \infty$). Besides, the derived bound converges at the rate of $O(\frac{1}{m})$ instead of $O(\frac{1}{\sqrt{m}})$ in (Pentina & Lampert, 2014; Amit & Meir, 2018), due to the use of Catoni's bound.

# 4 PAC-BAYES LOCALIZED META-LEARNING

## 4.1 OVERALL FRAMEWORK

Our motivation stems from a core challenge in PAC-Bayes meta-learning bound in 41, wherein the complexity term $\sum_{i=1}^{n} \frac{c_1' c_2'}{2c_2 n m_i \sigma_{\mathbf{w}}^2} \|\mathbb{E}\mathbf{w}_i^Q - \mathbf{w}^{\mathcal{Q}}\|^2$ is typically vital to the bound and so finding the tightest possible bound generally depends on minimizing this term. It is obvious that the optimal $\mathbf{w}^{\mathcal{Q}}$ is $\sum_{i=1}^{n} \frac{c_1' c_2' \mathbb{E}\mathbf{w}_i^Q}{2c_2 n m_i \sigma_{\mathbf{w}}^2}$. However, if the learned posteriors for each task are mutually exclusive, i.e., one learned posterior has a negative effect on another task, this term could be inevitably large.

$\mathbf{w}^{\mathcal{Q}}$ is the mean of hyperposterior $\mathcal{Q}$ and this term naturally indicates the divergence between the mean of prior $\mathbf{w}_i^P$ sampled from the hyperposterior $\mathcal{Q}$ and the mean of posterior $\mathbf{w}_i^Q$ in each task. Therefore, we propose to adaptively choose the mean of prior $\mathbf{w}_i^P$ according to task $i$. It is obvious that the complexity term vanishes if we set $\mathbf{w}_i^P = \mathbf{w}_i^Q$, but the prior $P_i$ in each task has to be chosen independently of the sample set $S_i$. Fortunately, the PAC-Bayes theorem allows us to choose prior upon the data distribution $D_i$. Therefore, we propose a prior predictor $\Phi : D^m \to \mathbf{w}^P$ which receives task data distribution $D^m$ and outputs the mean of prior $\mathbf{w}^P$. In this way, the generated priors could focus locally on those regions of model parameters that are of particular interest in solving specific tasks.

Particularly, the prior predictor is parameterized as $\Phi_{\mathbf{v}}$ with $\mathbf{v} \in \mathbb{R}^{d_{\mathbf{v}}}$. We abuse notation $\mathcal{P}$ and $\mathcal{Q}$ and assume $\mathbf{v}$ as a random variable distributed first according to the hyperprior $\mathcal{P}$, which we reformulate as $\mathcal{N}(0, \sigma_{\mathbf{v}}^2 I_{d_{\mathbf{v}}})$, and later according to hyperposterior $\mathcal{Q}$, which we reformulate as $\mathcal{N}(\mathbf{v}^{\mathcal{Q}}, \sigma_{\mathbf{v}}^2 I_{d_{\mathbf{v}}})$. Given a new task $i$, we first sample $\mathbf{v}$ from hyperposterior $\mathcal{N}(\mathbf{v}^{\mathcal{Q}}, \sigma_{\mathbf{v}}^2 I_{d_{\mathbf{v}}})$ and estimate the mean of prior $\mathbf{w}_i^P$ by leveraging prior predictor $\mathbf{w}_i^P = \Phi_{\mathbf{v}}(D_i^m)$. Then, the base learner utilizes the sample set $S_i$ and the prior $P_i = \mathcal{N}(\mathbf{w}_i^P, \sigma_{\mathbf{w}}^2 I_{d_{\mathbf{w}}})$ to produce a mean posterior $\mathbf{w}_i^Q = A_b(S_i, P_i)$, as illustrated in Figure 1(right).

## 4.2 LCC-BASED PRIOR PREDICTOR

To make $\mathbf{w}^P$ close to $\mathbf{w}^Q$ in each task, the prior predictor is required to (i) uncover the tight relationship between the sample set and model parameter. Intuitively, features and parameters yield similar local and global structures in their respective spaces in the classification problem. Features in the

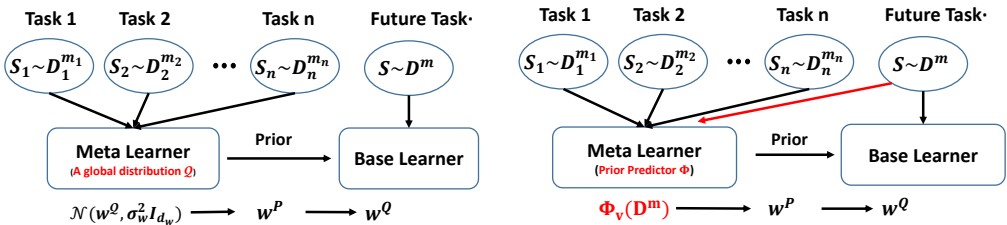

Figure 1: Comparison between meta-learning (left) and localized meta-learning (right). In regular meta-learning, the mean of prior $\mathbf{w}^P$ is sampled from a global hyperposterior distribution $\mathcal{Q} = \mathcal{N}(\mathbf{w}^{\mathcal{Q}}, \sigma_{\mathbf{w}}^2 I_{d_{\mathbf{w}}})$. In the localized meta-learning, $\mathbf{w}^P$ is produced by a prior predictor $\Phi_{\mathbf{v}}(D^m)$.

same category tend to be spatially clustered together while maintaining the separation between different classes. Take linear classifiers as an example, let $\mathbf{w}_k$ be the parameters w.r.t. category $k$, the separability between classes is implemented as $\mathbf{x} \cdot \mathbf{w}_k$, which also explicitly encourages intra-class compactness. A reasonable choice of $\mathbf{w}_k$ is to maximize the inner product distance with the input features in the same category and minimize the distance with the input features of the non-belonging categories. Besides, the prior predictor should be (ii) category-agnostic since it will be used continuously as new tasks and hence new categories become available. Lastly, it should be (iii) invariant under permutations of its inputs.

To satisfy the above conditions, we follow the idea of *nearest class mean classifier* (Mensink et al., 2013), which represents class parameter by averaging its feature embeddings. This idea has been explored in transductive few-shot learning problem (Bertinetto et al., 2016; Yang et al., 2018). Snell et al. (2017) learns a metric space across tasks such that when represented in this embedding, prototype (centroid) of each class can be used for label prediction in the new task. Qiao et al. (2018) directly predicts the classifier weights using the activations by exploiting the close relationship between the parameters and the activations in a neural network associated with the same category. In summary, the classification problem of each task is transformed as a generic metric learning problem which is shared across tasks. Once this mapping has been learned on observed tasks, due to the structure-preserving property, it could be easily generalized to new tasks. Formally, let each task be a $K$-class classification problem. Then the parameter of the classifier in task $i$ is represented as $\mathbf{w}_i = [\mathbf{w}_i[1], \ldots, \mathbf{w}_i[k], \ldots, \mathbf{w}_i[K]]$. The prior predictor for class $k$ could be defined as:

$$\mathbf{w}_i^P[k] = \Phi_{\mathbf{v}}(D_{ik}^{m_{ik}}) = \mathop{\mathbb{E}}_{S_{ik} \sim D_{ik}^{m_{ik}}} \frac{1}{m_{ik}} \sum_{\mathbf{x}_j \in S_{ik}} \phi_{\mathbf{v}}(\mathbf{x}_j), \tag{5}$$

where $\phi_{\mathbf{v}}(\cdot) : \mathbb{R}^d \to \mathbb{R}^{d_{\mathbf{w}}}$ is the feature embedding function, $m_{ik}$ is the number of samples belonging to category $k$, $S_{ik}$ and $D_{ik}$ are the sample set and data distribution for category $k$ in task $i$. We call this function the *expected prior predictor*. Since data distribution $D_{ik}$ is considered unknown and our only insight as to $D_{ik}$ is through the sample set $S_{ik}$, we approximate the expected prior predictor by its empirical counterpart, based on $m_{ik}$ observed samples in the category $k$:

$$\hat{\mathbf{w}}_i^P[k] = \hat{\Phi}_{\mathbf{v}}(S_{ik}) = \frac{1}{m_{ik}} \sum_{\mathbf{x}_j \in S_{ik}} \phi_{\mathbf{v}}(\mathbf{x}_j), \tag{6}$$

which we call the *empirical prior predictor*. Although we can implement the embedding function $\phi_{\mathbf{v}}(\cdot)$ with a multilayer perceptron (MLP), both input $\mathbf{x}$ and model parameter $\mathbf{w}$ are high-dimensional, making the empirical prior predictor $\hat{\Phi}_{\mathbf{v}}(\cdot)$ difficult to learn. According to Definition (3), any points on the latent manifold can be approximated by a linear combination of a set of anchor points. Inspired by this, if the anchor points are sufficiently localized, the empirical prior predictor $\hat{\Phi}_{\mathbf{v}}(S)$ can also be approximated by a linear function w.r.t. a set of codings. Accordingly, we propose an LCC-based prior predictor, which is defined as:

$$\bar{\mathbf{w}}_i^P[k] = \bar{\Phi}_{\mathbf{v}}(S_{ik}) = \frac{1}{m_{ik}} \sum_{\mathbf{x}_j \in S_{ik}} \sum_{\mathbf{u} \in C} \gamma_{\mathbf{u}}(\mathbf{x}_j) \Phi_{\mathbf{v}}(\mathbf{u}), \tag{7}$$

where $\Phi_{\mathbf{v}}(\mathbf{u}) \in \mathbb{R}^{d_{\mathbf{w}}}$ is the feature embedding of base $\mathbf{u} \in \mathbb{R}^d$. As such, the parameters of LCC-based prior predictor w.r.t. category $k$ can be represented as $\mathbf{v}_k = [\Phi_{\mathbf{v}_k}(\mathbf{u}_1), \Phi_{\mathbf{v}_k}(\mathbf{u}_2), \ldots, \Phi_{\mathbf{v}_k}(\mathbf{u}_{|C|})]$. Lemma 1 illustrates the approximation error.

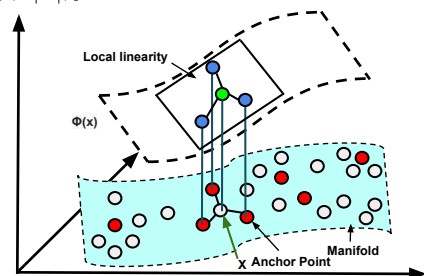

Figure 2: A geometric view of Local Coordinate Coding. Given a set of anchor points, if data lie on a manifold, the empirical prior predictor $\hat{\Phi}_{\mathbf{v}}(S)$ can be locally approximated by a linear function w.r.t. the coding. Given all bases, $\hat{\Phi}_{\mathbf{v}}(S)$ can be globally approximated.

**Lemma 1.** *(Empirical Pior Predictor Approximation) Given the definition of $\hat{\mathbf{w}}_i^P[k]$ and $\bar{\mathbf{w}}_i^P[k]$ in Eq. (6) and Eq. (7), let $(\gamma, C)$ be an arbitrary coordinate coding on $\mathbb{R}^d$ and $\phi$ be an $(\alpha, \beta)$-Lipschitz smooth function. We have for all $\mathbf{x} \in \mathbb{R}^d$*

$$\|\hat{\mathbf{w}}_i^P[k] - \bar{\mathbf{w}}_i^P[k]\| \le \frac{1}{m_{ik}} \sum_{\mathbf{x}_j \in S_{ik}} \left( \alpha \|\mathbf{x}_j - \bar{\mathbf{x}}_j\| + \beta \sum_{\mathbf{u} \in C} \|\bar{\mathbf{x}}_j - \mathbf{u}\|^2 \right) = O_{\alpha,\beta}(\gamma, C), \quad (8)$$

*where $\bar{\mathbf{x}}_j = \sum_{\mathbf{u} \in C} \gamma_{\mathbf{u}}(\mathbf{x}_j)\mathbf{u}$. Then given any $\epsilon > 0$, there exists a coding $(\gamma, C)$ such that*

$$\begin{aligned} |C| &\le (1 + d_{\mathcal{M}})\mathcal{N}(\epsilon, \mathcal{M}), \\ O_{\alpha,\beta}(\gamma, C) &\le [\alpha c_{\mathcal{M}} + (1 + 5\sqrt{d_{\mathcal{M}}})\beta]\epsilon^2. \end{aligned} \quad (9)$$

*Proof.* See appendix B.1 for the proof.

The first inequality of Lemma 1 demonstrates that a good LCC-based prior predictor should make $\mathbf{x}$ close to its physical approximation $\bar{\mathbf{x}}$ and should be localized. The second and third inequality show that if a set of anchor points $C$ has cardinality $O(d_{\mathcal{M}}\mathcal{N}(\epsilon, \mathcal{M}))$, emprical prior predictor can be linearly approximated using LCC up to accuracy $O(\sqrt{d_{\mathcal{M}}}\epsilon^2)$. The complexity of the LCC coding scheme depends only on the number of anchor points $|C|$ instead of the input dimension. In fact, a small $|C|$ is usually sufficient to achieve good approximation.

**Optimization of LCC.** We minimize the first inequality in (8) to obtain a set of anchor points. As with (Yu et al., 2009), we simplify the localization error term by assuming $\bar{\mathbf{x}} = \mathbf{x}$, and then we optimize the following objective function:

$$\arg\min_{\gamma, C} \sum_{i=1}^n \sum_{\mathbf{x}_j \in S_i} \alpha \|\mathbf{x}_j - \bar{\mathbf{x}}_j\|^2 + \beta \sum_{\mathbf{u} \in C} \|\mathbf{x}_j - \mathbf{u}\|^2 \qquad s.t. \quad \sum_{\mathbf{u} \in C} \gamma_{\mathbf{u}}(\mathbf{x}) = 1, \forall \mathbf{x}, \quad (10)$$

where $\bar{\mathbf{x}} = \sum_{\mathbf{u} \in C} \gamma_{\mathbf{u}}(\mathbf{x})\mathbf{u}$. In practice, we update $C$ and $\gamma$ by alternately optimizing a LASSO problem and a least-square regression problem, respectively.

### 4.3 PAC-Bayes Localized Meta-Learning Bound with Gaussian Randomization

In order to derive a PAC-Bayes generalization bound for localized meta-learning, we first bound the approximation error between expected prior predictor and LCC-based prior predictor.

**Lemma 2.** *Given the definition of $\mathbf{w}^P$ and $\bar{\mathbf{w}}^P$ in Eq. (5) and (7), let $\mathcal{X}$ be a compact set with radius $R$, i.e., $\forall \mathbf{x}, \mathbf{x}' \in \mathcal{X}, \|\mathbf{x} - \mathbf{x}'\| \le R$. For any $\delta \in (0, 1]$ with probability $\ge 1 - \delta$, we have*

$$\|\mathbf{w}^P - \bar{\mathbf{w}}^P\|^2 \le \sum_{k=1}^K \left( \frac{\alpha R}{\sqrt{m_{ik}}} (1 + \sqrt{\frac{1}{2}\log(\frac{1}{\delta})}) + O_{\alpha,\beta}(\gamma, C) \right)^2. \quad (11)$$

*Proof.* See appendix B.2 for the proof.

Lemma 2 shows that the approximation error between expected prior predictor and LCC-based prior predictor depends on (i) the concentration of prior predictor and (ii) the quality of LCC coding scheme. The first term implies the number of samples for each category should be larger for better approximation. This is consistent with the results of estimating the center of mass (Cristianini & Shawe-Taylor, 2004). Based on Lemma 2, we have the following PAC-Bayes LML bound.

**Theorem 3.** *Consider the Localized Meta-Learning (LML) framework, give the hyperprior $\mathcal{P} = \mathcal{N}(0, \sigma_{\mathbf{v}}^2 I_{d_{\mathbf{v}}})$, then for any hyperposterior $\mathcal{Q}$, any $c_1, c_2 > 0$ and any $\delta \in (0, 1]$ with probability $\geq 1 - \delta$ we have,*

$$er(\mathcal{Q}) \leq c_1' c_2' \hat{er}(\mathcal{Q}) + (\sum_{i=1}^{n} \frac{c_1' c_2'}{2c_2 n m_i \sigma_{\mathbf{v}}^2} + \frac{c_1'}{2c_1 n \sigma_{\mathbf{v}}^2}) \|\mathbf{v}^{\mathcal{Q}}\|^2 + \sum_{i=1}^{n} \frac{c_1' c_2'}{c_2 n m_i \sigma_{\mathbf{w}}^2} \|\mathbb{E}_{\mathbf{v}} \mathbf{w}_i^Q - \bar{\Phi}_{\mathbf{v}^{\mathcal{Q}}}(S_i)\|^2$$

$$+ \sum_{i=1}^{n} \frac{c_1' c_2'}{c_2 n m_i \sigma_{\mathbf{w}}^2} \left( \frac{1}{\sigma_{\mathbf{w}}^2} \sum_{k=1}^{K} \left( \frac{\alpha R}{\sqrt{m_{ik}}} (1 + \sqrt{\frac{1}{2} \log(\frac{4n}{\delta})}) + O_{\alpha, \beta}(\gamma, C) \right)^2 + d_{\mathbf{w}} K (\frac{\sigma_{\mathbf{v}}}{\sigma_{\mathbf{w}}})^2 \right)$$

$$+ \sum_{i=1}^{n} \frac{c_1' c_2'}{c_2 n m_i \sigma_{\mathbf{w}}^2} \log \frac{4n}{\delta} + \frac{c_1'}{2c_1 n \sigma_{\mathbf{v}}^2} \log \frac{2}{\delta}, \tag{12}$$

*where $c_1' = \frac{c_1}{1 - e^{-c_1}}$ and $c_2' = \frac{c_2}{1 - e^{-c_2}}$. To get a better understanding, we further simplify the notation and obtain that*

$$er(\mathcal{Q}) \leq c_1' c_2' \hat{er}(\mathcal{Q}) + (\sum_{i=1}^{n} \frac{c_1' c_2'}{2c_2 n m_i \sigma_{\mathbf{v}}^2} + \frac{c_1'}{2c_1 n \sigma_{\mathbf{v}}^2}) \|\mathbf{v}^{\mathcal{Q}}\|^2 + \sum_{i=1}^{n} \frac{c_1' c_2'}{c_2 n m_i \sigma_{\mathbf{w}}^2} \|\mathbb{E}_{\mathbf{v}} \mathbf{w}_i^Q - \bar{\Phi}_{\mathbf{v}^{\mathcal{Q}}}(S_i)\|^2$$

$$+ const(\alpha, \beta, R, \delta, n, m_i). \tag{13}$$

*Proof.* See appendix B.3 for the proof.

Similarly with the PAC-Bayes meta-learning bound in Theorem 2 and the bounds in (Pentina & Lampert, 2014; Amit & Meir, 2018), the expected task error $er(\mathcal{Q})$ is bounded by the empirical task error $\hat{er}(\mathcal{Q})$ plus the task-complexity and environment-complexity terms. The main innovation here is to exploit the potential to choose the mean of prior $\mathbf{w}^P$ based on task data $S$. Intuitively, if the selection of the LCC-based prior predictor is appropriate, it will narrow the divergence between the mean of prior $\mathbf{w}_i^P$ sampled from the hyperposterior $\mathcal{Q}$ and the mean of posterior $\mathbf{w}_i^Q$ in each task. Therefore, the bound can be tighter than the ones in the meta-learning framework. Our empirical study in Section 5 illustrates that the algorithms derived from this bound can achieve better performance than the methods derived from standard PAC-Bayes meta-learning bounds.

When one is choosing the LCC-based prior predictor $\bar{\Phi}_{\mathbf{v}}(\cdot)$, the number of anchor points $|C|$, there is a balance between accuracy and simplicity. As we increase $|C|$, it will essentially increase the expressive power of $\bar{\Phi}_{\mathbf{v}}(\cdot)$ and reduce the complexity term $\|\mathbb{E}_{\mathbf{v}} \mathbf{w}^Q - \bar{\Phi}_{\mathbf{v}^{\mathcal{Q}}}(S)\|^2$. However, at the same time, it will increase the complexity term $\|\mathbf{v}^{\mathcal{Q}}\|^2$ and make the bound loose. If we set $|C|$ to 1, it is degraded to the regular meta-learning framework.

### 4.4 LOCALIZED META-LEARNING ALGORITHM

Since the bound in (27) holds uniformly w.r.t. $\mathcal{Q}$, the guarantees of Theorem 3 also hold for the resulting learned hyperposterior $\mathcal{Q} = \mathcal{N}(\mathbf{v}^{\mathcal{Q}}, \sigma_{\mathbf{v}}^2 I_{d_{\mathbf{v}}})$, so the mean of prior $\mathbf{w}^P$ sampled from the learned hyperposterior work well for future tasks. The PAC-Bayes localized meta-learning bound in (27) can be compactly written as

$$\sum_{i=1}^{n} \mathbb{E}_{\mathbf{v}} \hat{er}_i(Q_i = A_b(S_i, P)) + \alpha_1 \|\mathbf{v}^{\mathcal{Q}}\|^2 + \sum_{i=1}^{n} \frac{\alpha_2}{m_i} \|\mathbb{E}_{\mathbf{v}} \mathbf{w}_i^Q - \bar{\Phi}_{\mathbf{v}^{\mathcal{Q}}}(S_i)\|^2, \tag{14}$$

where $\alpha_1, \alpha_2 > 0$ are hyperparameters. For task $i$, the learning algorithm $A_b(\cdot)$ can be formulated as $\mathbf{w}_i^\star = \arg\min_{\mathbf{w}_i^Q} \mathbb{E}_{\mathbf{v}} \hat{er}_i(Q_i = \mathcal{N}(\mathbf{w}_i^Q, \sigma_{\mathbf{w}}^2 I_{d_{\mathbf{w}}}))$. Following Amit & Meir (2018), we jointly opti-

mize the parameters of LCC-based prior predictor $\mathbf{v}$ and the parameters of classifiers in each task $\mathbf{w}_1, \mathbf{w}_2, \ldots, \mathbf{w}_n$, which is formulated as

$$\arg \min_{\mathbf{v}, \mathbf{w}_1, \ldots, \mathbf{w}_n} \sum_{i=1}^n \mathbb{E}_{\mathbf{v}} \hat{er}_i(\mathbf{w}_i) + \alpha_1 \|\mathbf{v}^{\mathcal{Q}}\|^2 + \sum_{i=1}^n \frac{\alpha_2}{m_i} \|\mathbb{E}_{\mathbf{v}} \mathbf{w}_i^Q - \bar{\Phi}_{\mathbf{v}^{\mathcal{Q}}}(S_i)\|^2. \tag{15}$$

We can optimize $\mathbf{v}$ and $\mathbf{w}$ via mini-batch SGD. The details of algorithms for meta-training are given in Algorithms 1. The expectation over Gaussian distribution and its gradient can be efficiently

---

**Algorithm 1** Localized Meta-Learning (LML) algorithm

**Input:** Data sets of observed tasks: $S_1, \ldots, S_n$.
**Output:** Learned prior predictor $\bar{\Phi}$ parameterized by $\mathbf{v}$.
Initialize $\mathbf{v} \in \mathbb{R}^{d_{\mathbf{v}}}$ and $\mathbf{w}_i \in \mathbb{R}^{d_{\mathbf{w}}}$ for $i = 1 \ldots, n$.
Construct LCC scheme $(\gamma, C)$ from the whole training data by optimizing Eq. (10).
**while** not converged **do**
    **for** each task $i \in \{1, \ldots, n\}$ **do**
        Sample a random mini-batch from the data $S_i' \subset S_i$.
        Approximate $\mathbb{E}_{\mathbf{v}} \hat{er}_i(\mathbf{w}_i)$ using $S_i'$.
    **end for**
    Compute the objective in (15), i.e. $J \leftarrow \sum_{i=1}^n \mathbb{E}_{\mathbf{v}} \hat{er}_i(\mathbf{w}_i) + \alpha_1 \|\mathbf{v}^{\mathcal{Q}}\|^2 + \sum_{i=1}^n \frac{\alpha_2}{m_i} \|\mathbb{E}_{\mathbf{v}} \mathbf{w}_i^Q - \bar{\Phi}_{\mathbf{v}^{\mathcal{Q}}}(S_i)\|^2$.
    Evaluate the gradient of $J$ w.r.t. $\{\mathbf{v}, \mathbf{w}_1, \ldots, \mathbf{w}_n\}$ using backpropagation.
    Take an optimization step.
**end while**

---

estimated by using the re-parameterization trick (Kingma & Welling, 2014; Rezende et al., 2014). For example, to sample $\mathbf{w}$ from the posterior $Q = \mathcal{N}(\mathbf{w}^Q, \sigma_{\mathbf{w}}^2 I_{d_{\mathbf{w}}})$, we first draw $\xi \sim \mathcal{N}(0, I_{d_{\mathbf{w}}})$ and then apply the deterministic function $\mathbf{w}^Q + \xi \odot \sigma$, where $\odot$ is an element-wise multiplication.

## 5 EXPERIMENTS

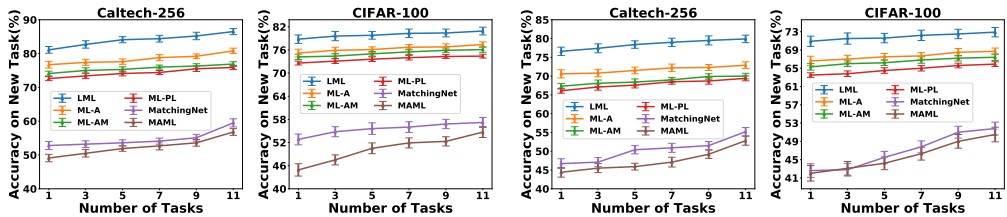

(a) With pre-trained feature extractor      (b) Without pre-trained feature extractor

Figure 3: The average test accuracy of learning a new task for different number of training tasks ($|C| = 64$).

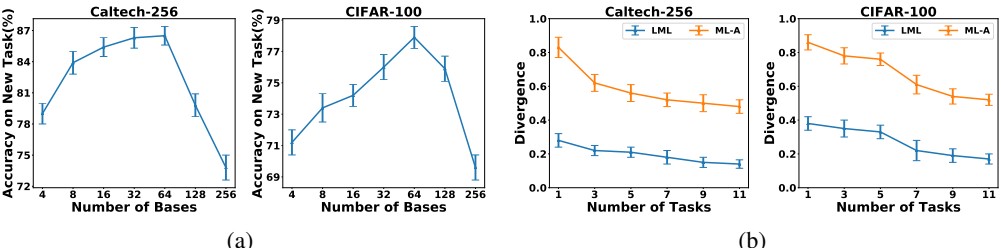

(a)          (b)

Figure 4: (a) The impact of the number of bases $|C|$ in LCC. (b) The divergence value (normalized) between the mean generated prior $\mathbf{w}^P$ and the mean of learned posterior $\mathbf{w}^Q$.

### 5.1 DATASETS AND SETUP

We use CIFAR-100 and Caltech-256 in our experiments. CIFAR-100 (Krizhevsky, 2009) contains 60,000 images from 100 fine-grained categories and 20 coarse-level categories. As in (Zhou et al.,

2018), we use 64, 16, and 20 classes for meta-training, meta-validation, and meta-testing, respectively. Caltech-256 has 30,607 color images from 256 classes (Griffin et al., 2007). Similarly, we split the dataset into 150, 56 and 50 classes for meta-training, meta-validation, and meta-testing. We consider 5-way classification problem. Each task is generated by randomly sampling 5 categories and each category contains 50 samples. The base model uses the convolutional architecture in (Finn et al., 2017), which consists of 4 convolutional layers, each with 32 filters and a fully-connected layer mapping to the number of classes on top. High dimensional data often lies on some low dimensional manifolds. We utilize an auto-encoder to extract the semantic information of image data and then construct the LCC scheme based on the embeddings. The parameters of prior predictor and base model are random perturbations in the form of Gaussian distribution. We design two different meta-learning environment setting to validate the efficacy of the proposed method. The first one uses the pre-trained base model as an initialization, which utilizes all the meta-training classes (64-class classification in CIFAR-100 case) to train the feature extractor. The second one uses the random initialization. We compare the proposed **LML** method with **ML-PL** method (Pentina & Lampert, 2014), **ML-AM** method (Amit & Meir, 2018) and **ML-A** which is derived from Theorem 2. In these methods, we use their main theorems about the PAC-Bayes generalization bound to derive the objective for the algorithm. We also compare with two typical meta-learning few-shot learning methods: MAML (Finn et al., 2017) and MatchingNet (Vinyals et al., 2016). To ensure a fair comparison, all approaches adopt the same network architecture and pre-trained feature extractor.

## 5.2 RESULTS

In Figure 3, we demonstrate the average test error of learning a new task based on the number of training tasks in different settings (with or without a pre-trained feature extractor). It is obvious that the performance continually increases as we increase the number of training tasks for all the methods. This is consistent with the generalization bounds that the complexity term converges to zero if large numbers of tasks are observed. ML-A consistently outperforms ML-PL and ML-AM since the bound w.r.t. ML-A in Theorem 2 converges at the rate of $O(\frac{1}{m})$ while the bounds w.r.t. ML-PL and ML-AM converge at the rate of $O(\frac{1}{\sqrt{m}})$. This demonstrates the importance of using tight generalization bound. Our proposed LML significantly outperforms the baselines, which validates the effectiveness of the proposed LCC-based prior predictor. It is a more suitable representation for meta-knowledge than the traditional global hyperposterior in ML-A, ML-AM, and ML-PL.

Moreover, we can find that all PAC-Bayes baselines outperform MAML and MatchingNet. Note that MAML and MatchingNet adopt the episodic training paradigm to solve the few-shot learning problem. The meta-training process requires millions of tasks and each task contains limited samples, which is not the case in our experiment. Scarce tasks in meta-training leads to severely meta-overfitting. In our method, the learned prior serves both as an initialization of base model and as a regularizer which restricts the solution space while allowing variation based on specific task data. It yields a model with smaller error than its unbiased counterpart when applied to a similar task.

Finally, we observe that if the pre-trained feature extractor is provided, all of these methods do better than meta-training with random initialization. This is because pre-trained feature extractor can be regarded as a data-dependent hyperpior. It is closer to the hyperposteior than the randomly initialized hyperprior. Therefore, it reduces the environment complexity term and improves the generalization performance.

In Figure 4(b), we show the divergence between the mean of generated prior $\mathbf{w}^P$ from meta model and the mean of learned posterior $\mathbf{w}^Q$ for LML and ML-A. This further validates the effectiveness of the LCC-based prior predictor which could narrow the divergence term and thus tight the bound. In Figure 4(a), we vary the number of bases $|C|$ in LCC scheme from $4$ to $256$, the optimal value is around $64$ in both datasets. This indicates that LML is sensitive to the number of bases $|C|$, which further affects the quality of LCC-based prior predictor and the performance of LML.

## 6 RELATED WORK

**Meta-Learning.** Meta-learning literature commonly considers the empirical task error by directly optimizing a loss of meta learner across tasks in the training data. Recently, this has been success-

fully applied in a variety of models for few-shot learning (Ravi & Larochelle, 2017; Snell et al., 2017; Finn et al., 2017; Vinyals et al., 2016). Although Vuorio et al. (2018); Rusu et al. (2019); Zintgraf et al. (2019); Wang et al. (2019) consider task adaptation when using meta-knowledge for specific tasks, all of them are not based on generalization error bounds, which is the focus of our work. Meta-learning in the online setting has regained attention recently (Denevi et al., 2018b;a; 2019; Balcan et al., 2019), in which online-to-batch conversion results could imply generalization bounds. Galanti et al. (2016) analyzes transfer learning in neural networks with PAC-Bayes tools. Most related to our work are (Pentina & Lampert, 2014; Amit & Meir, 2018) which provide a PAC-Bayes generalization bound for meta-learning framework. In contrast, neither work considers localized meta-knowledge for specific tasks.

**Localized PAC-Bayes Learning.** There has been a prosperous line of research for learning priors to improve the PAC-Bayes bounds Catoni (2007); Guedj (2019). (Parrado-Hernández et al., 2012) showed that priors can be learned by splitting the available training data into two parts, one for learning the prior, one for learning the posterior. (Lever et al., 2013) derived an expression for the overall best prior, i.e. the distribution resulting in the smallest possible bound value and bounded the KL divergence by a term independent of data distribution. Recently, (Rivasplata et al., 2018) bounded the KL divergence by investigating the stability of the hypothesis. (Dziugaite & Roy, 2018) optimized the prior term in a differentially private way. In summary, theses methods construct some quantities that reflect the underlying data distribution, rather than the sample set, and then choose the prior $P$ based on these quantities. These works, however, are only applicable for single-task problem and could not transfer knowledge across tasks in meta-learning setting.

## 7 CONCLUSION

This work contributes a novel localized meta-learning framework from a theoretical perspective. We propose a generalization bound based on PAC-Bayes theory with Gaussian randomization. Instead of formulating meta-knowledge as a global distribution, we propose an LCC-based prior predictor to output local meta-knowledge by using task information. We further develop a practical algorithm with deep neural network based on the bound. An interesting topic for future work would be to explore other principle to construct the prior predictor and apply the localized meta-learning framework to a more realistic scenario that tasks are sampled non-i.i.d. from an environment. Another challenging problem is to extend our techniques to derive localized meta-learning algorithms for regression and reinforcement learning problem.

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

This supplementary document contains the technical proofs of theoretical results and details of experiments. It is structured as follows: Appendix A present notations for prior predictor. Appendix B gives the proofs of the main results. Appendix B.1 and B.2 show the approximation error between LCC-based prior predictor and empirical prior predictor, expected prior predictor, respectvely. They are used in the proof of Theorem 3. Next, in Appendix B.3 and B.4 we show the PAC-Bayes generalization bound of localized meta-learning in Theorem 3 and also provides the PAC-Bayes generalization bound of regular meta-learning in Theorem 2. Finally, details of experiments are presented in Appendix C.

## A    NOTATIONS

Let $\phi_{\mathbf{v}}(\cdot) : \mathbb{R}^d \rightarrow \mathbb{R}^{d_{\mathbf{w}}}$ be the feature embedding function. $m_{ik}$ denotes the number of samples belonging to category $k$. $S_{ik}$ and $D_{ik}$ are the sample set and data distribution for category $k$ in task $i$, respectively. Then, the expected prior predictor w.r.t. class $k$ in task $i$ is defined as:

$$\mathbf{w}_i^P[k] = \Phi_{\mathbf{v}}(D_{ik}^{m_{ik}}) = \mathop{\mathbb{E}}_{S_{ik} \sim D_{ik}^{m_{ik}}} \frac{1}{m_{ik}} \sum_{\mathbf{x}_j \in S_{ik}} \phi_{\mathbf{v}}(\mathbf{x}_j).$$

The empirical prior predictor w.r.t. class $k$ in task $i$ is defined as:

$$\hat{\mathbf{w}}_i^P[k] = \hat{\Phi}_{\mathbf{v}}(S_{ik}) = \frac{1}{m_{ik}} \sum_{\mathbf{x}_j \in S_{ik}} \phi_{\mathbf{v}}(\mathbf{x}_j).$$

The LCC-based prior predictor w.r.t. class $k$ in task $i$ is defined as:

$$\bar{\mathbf{w}}_i^P[k] = \bar{\Phi}_{\mathbf{v}}(S_{ik}) = \frac{1}{m_{ik}} \sum_{\mathbf{x}_j \in S_{ik}} \sum_{\mathbf{u} \in C} \gamma_{\mathbf{u}}(\mathbf{x}_j) \Phi_{\mathbf{v}}(\mathbf{u}).$$

## B    THEORETICAL RESULTS

### B.1    PROOF OF LEMMA 1

This lemma bounds the error between the empirical prior predictor $\hat{\mathbf{w}}_i^P[k]$ and the LCC-based prior predictor $\bar{\mathbf{w}}_i^P[k]$.

**Lemma 1** Given the definition of $\hat{\mathbf{w}}_i^P[k]$ and $\bar{\mathbf{w}}_i^P[k]$ in Eq. (6) and Eq. (7), let $(\gamma, C)$ be an arbitrary coordinate coding on $\mathbb{R}^{d_x}$ and $\phi$ be an $(\alpha, \beta)$-Lipschitz smooth function and . We have for all $\mathbf{x} \in \mathbb{R}^{d_x}$

$$\|\hat{\mathbf{w}}_i^P[k] - \bar{\mathbf{w}}_i^P[k]\| \leq \frac{1}{m_{ik}} \sum_{\mathbf{x}_j \in S_{ik}} \left( \alpha\|\mathbf{x}_j - \bar{\mathbf{x}}_j\| + \beta \sum_{\mathbf{u} \in C} \|\bar{\mathbf{x}}_j - \mathbf{u}\|^2 \right) = O_{\alpha,\beta}(\gamma, C), \quad (16)$$

where $\bar{\mathbf{x}}_j = \sum_{\mathbf{u} \in C} \gamma_{\mathbf{u}}(\mathbf{x}_j)\mathbf{u}$. Then given any $\epsilon > 0$, there exists a coding $(\gamma, C)$ such that

$$|C| \leq (1 + d_{\mathcal{M}})\mathcal{N}(\epsilon, \mathcal{M}),$$
$$O_{\alpha,\beta}(\gamma, C) \leq [\alpha c_{\mathcal{M}} + (1 + 5\sqrt{d_{\mathcal{M}}})\beta]\epsilon^2. \quad (17)$$

*Proof.* Let $\bar{\mathbf{x}}_j = \sum_{\mathbf{u} \in C} \gamma_{\mathbf{u}}(\mathbf{x}_j)\mathbf{u}$. We have

$$\|\hat{\Phi}_{\mathbf{v}}(S_{ik}) - \bar{\Phi}_{\mathbf{v}}(S_{ik})\|_2$$

$$= \frac{1}{m_{ik}} \sum_{\mathbf{x}_j \in S_{ik}} \|\Phi_{\mathbf{v}}(\mathbf{x}_j) - \sum_{\mathbf{u} \in C} \gamma_{\mathbf{u}}(\mathbf{x}_j)\Phi_{\mathbf{v}}(\mathbf{u})\|_2$$

$$\leq \frac{1}{m_{ik}} \sum_{\mathbf{x}_j \in S_{ik}} \left( \|\Phi_{\mathbf{v}}(\mathbf{x}_j) - \Phi_{\mathbf{v}}(\bar{\mathbf{x}}_j)\|_2 + \| \sum_{\mathbf{u} \in C} \gamma_{\mathbf{u}}(\mathbf{x}_j)(\Phi_{\mathbf{v}}(\mathbf{u}) - \Phi_{\mathbf{v}}(\bar{\mathbf{x}}_j)\|_2 \right)$$

$$= \frac{1}{m_{ik}} \sum_{\mathbf{x}_j \in S_{ik}} \left( \|\Phi_{\mathbf{v}}(\mathbf{x}_j) - \Phi_{\mathbf{v}}(\bar{\mathbf{x}}_j)\|_2 + \| \sum_{\mathbf{u} \in C} \gamma_{\mathbf{u}}(\mathbf{x}_j)(\Phi_{\mathbf{v}}(\mathbf{u}) - \Phi_{\mathbf{v}}(\sum_{\mathbf{u} \in C} \gamma_{\mathbf{u}}(\mathbf{x}_j)\mathbf{u})) - \nabla\Phi_{\mathbf{v}}(\bar{\mathbf{x}}_j)(\mathbf{u} - \bar{\mathbf{x}}_j)\|_2 \right)$$

$$\leq \frac{1}{m_{ik}} \sum_{\mathbf{x}_j \in S_{ik}} \left( \|\Phi_{\mathbf{v}}(\mathbf{x}_j) - \Phi_{\mathbf{v}}(\bar{\mathbf{x}}_j)\|_2 + \sum_{\mathbf{u} \in C} |\gamma_{\mathbf{u}}(\mathbf{x}_j)| \|(\Phi_{\mathbf{v}}(\mathbf{u}) - \Phi_{\mathbf{v}}(\sum_{\mathbf{u} \in C} \gamma_{\mathbf{u}}(\mathbf{x}_j)\mathbf{u})) - \nabla\Phi_{\mathbf{v}}(\bar{\mathbf{x}}_j)(\mathbf{u} - \bar{\mathbf{x}}_j)\|_2 \right)$$

$$\leq \frac{1}{m_{ik}} \sum_{\mathbf{x}_j \in S_{ik}} \left( \alpha\|\mathbf{x}_j - \bar{\mathbf{x}}_j\|_2 + \beta \sum_{\mathbf{u} \in C} \|\bar{\mathbf{x}}_j - \mathbf{u}\|_2^2 \right) = O_{\alpha,\beta}(\gamma, C)$$

In the above derivation, the first inequality holds by the triangle inequality. The second equality holds since $\sum_{\mathbf{u} \in C} \gamma_{\mathbf{u}}(\mathbf{x}_j) = 1$ for all $\mathbf{x}_j$. The last inequality uses the assumption of $(\alpha, \beta)$-Lipschitz smoothness of $\Phi_{\mathbf{v}}(\cdot)$.

According to the Manifold Coding Theorem in (Yu et al., 2009), if the data points $\mathbf{x}$ lie on a compact smooth manifold $\mathcal{M}$. Then given any $\epsilon > 0$, there exists anchor points $C \subset \mathcal{M}$ and coding $\gamma$ such that

$$|C| \leq (1 + d_{\mathcal{M}})\mathcal{N}(\epsilon, \mathcal{M}),$$

$$\frac{1}{m_{ik}} \sum_{\mathbf{x}_j \in S_{ik}} \left( \alpha\|\mathbf{x}_j - \bar{\mathbf{x}}_j\|_2 + \beta \sum_{\mathbf{u} \in C} \|\bar{\mathbf{x}}_j - \mathbf{u}\|_2^2 \right) \leq [\alpha c_{\mathcal{M}} + (1 + 5\sqrt{d_{\mathcal{M}}})\beta]\epsilon^2. \quad (18)$$

This implies the desired bound. $\square$

The first inequality of this lemma demonstrates that the quality of LCC approximation is bounded by two terms: the first term $\|\mathbf{x}_j - \bar{\mathbf{x}}_j\|_2$ indicates $\mathbf{x}$ should be close to its physical approximation $\bar{\mathbf{x}}$, the second term $\|\bar{\mathbf{x}}_j - \mathbf{u}\|$ implies that the coding should be localized. The second and third inequality show that the approximation error of local coordinate coding depends on the intrinsic dimension of the manifold instead of the dimension of input. If a set of anchor points $C$ has cardinality $O(d_{\mathcal{M}}\mathcal{N}(\epsilon, \mathcal{M}))$, emprical prior predictor can be linearly approximated using LCC up to accuracy $O(\sqrt{d_{\mathcal{M}}}\epsilon^2)$.

## B.2 PROOF OF LEMMA 2

In order to proof Lemma 2, we first introduce a relevant theorem.

**Theorem 4.** *(Vector-valued extension of McDiarmid's inequality (Rivasplata et al., 2018)) Let* $\mathbf{X}_1, \ldots, \mathbf{X}_m \in \mathcal{X}$ *be independent random variables, and* $f : \mathcal{X}^m \to \mathbb{R}^{d_w}$ *be a vector-valued mapping function. If, for all* $i \in \{1, \ldots, m\}$, *and for all* $\mathbf{x}_1, \ldots, \mathbf{x}_m, \mathbf{x}_i' \in \mathcal{X}$, *the function* $f$ *satisfies*

$$\sup_{\mathbf{x}_i, \mathbf{x}_i'} \|f(\mathbf{x}_{1:i-1}, \mathbf{x}_i, \mathbf{x}_{i+1:m}) - f(\mathbf{x}_{1:i-1}, \mathbf{x}_i', \mathbf{x}_{i+1:m})\| \leq c_i \quad (19)$$

*Then* $\mathbb{E}\|f(\mathbf{X}_{1:m}) - \mathbb{E}[f(\mathbf{X}_{1:m})]\| \leq \sqrt{\sum_{i=1}^{m} c_i^2}$. *For any* $\delta \in (0, 1)$ *with probability* $\geq 1 - \delta$ *we have*

$$\|f(\mathbf{X}_{1:m}) - \mathbb{E}[f(\mathbf{X}_{1:m})]\| \leq \sqrt{\sum_{i=1}^{m} c_i^2} + \sqrt{\frac{\sum_{i=1}^{m} c_i^2}{2} \log(\frac{1}{\delta})}. \quad (20)$$

The above theorem indicates that bounded differences in norm implies the concentration of $f(\mathbf{X}_{1:m})$ around its mean in norm, i.e., $\|f(\mathbf{X}_{1:m}) - \mathbb{E}[f(\mathbf{X}_{1:m})]\|$ is small with high probability.

425 Then, we bound the error between expected prior predictor $\mathbf{w}_i^P$ and the empirical prior predictor
426 $\hat{\mathbf{w}}_i^P$.

**Lemma 3.** *Given the definition of $\mathbf{w}_i^P[k]$ and $\hat{\mathbf{w}}_i^P[k]$ in (5) and (6), let $\mathcal{X}$ be a compact set with radius $R$, i.e., $\forall \mathbf{x}, \mathbf{x}' \in \mathcal{X}, \|\mathbf{x} - \mathbf{x}'\| \le R$. For any $\delta \in (0, 1]$ with probability $\ge 1 - \delta$, we have*

$$\|\mathbf{w}_i^P[k] - \hat{\mathbf{w}}_i^P[k]\| \le \frac{\alpha R}{\sqrt{m_{ik}}}(1 + \sqrt{\frac{1}{2}\log(\frac{1}{\delta})}). \tag{21}$$

*Proof.* According to the definition of $\hat{\Phi}_{\mathbf{v}}(\cdot)$ in (6), for all points $\mathbf{x}_1, \ldots, \mathbf{x}_{j-1}, \mathbf{x}_{j+1}, \ldots, \mathbf{x}_{m_k}, \mathbf{x}_j'$ in the sample set $S_{ik}$, we have

$$\sup_{\mathbf{x}_i, \mathbf{x}_i'} \|\hat{\Phi}_{\mathbf{v}}(\mathbf{x}_{1:j-1}, \mathbf{x}_j, \mathbf{x}_{j+1:m_k}) - \hat{\Phi}_{\mathbf{v}}(\mathbf{x}_{1:j-1}, \mathbf{x}_j', \mathbf{x}_{j+1:m_k})\|$$

$$= \frac{1}{m_{ik}} \sup_{\mathbf{x}_j, \mathbf{x}_j'} \|\Phi_{\mathbf{v}}(\mathbf{x}_j) - \Phi_{\mathbf{v}}(\mathbf{x}_j')\| \le \frac{1}{m_{ik}} \sup_{\mathbf{x}_j, \mathbf{x}_j'} \alpha\|\mathbf{x}_j - \mathbf{x}_j'\| \le \frac{\alpha R}{m_{ik}}, \tag{22}$$

where $R$ denotes the domain of $\mathbf{x}$, say $R = \sup_{\mathbf{x}} \|\mathbf{x}\|$. The first inequality follows from the Lipschitz smoothness condition of $\Phi_{\mathbf{v}}(\cdot)$ and the second inequality follows by the definition of domain $\mathcal{X}$. Utilizing Theorem 4, for any $\delta \in (0, 1]$ with probability $\ge 1 - \delta$ we have

$$\|\mathbf{w}_i^P[k] - \hat{\mathbf{w}}_i^P[k]\| = \|\hat{\Phi}_{\mathbf{v}}(S_{ik}) - \mathbb{E}[\hat{\Phi}_{\mathbf{v}}(S_{ik})]\| \le \frac{\alpha R}{\sqrt{m_{ik}}}(1 + \sqrt{\frac{1}{2}\log(\frac{1}{\delta})}). \tag{23}$$

427 This implies the bound. $\qquad\square$

428 Lemma 3 shows that the bounded difference of function $\Phi_{\mathbf{v}}(\cdot)$ implies its concentration, which can
429 be further used to bound the differences between empirical prior predictor $\bar{\mathbf{w}}_i^P[k]$ and expected prior
430 predictor $\mathbf{w}_i^P[k]$. Now, we bound the error between expected prior predictor $\mathbf{w}_i^P$ and the LCC-based
431 prior predictor $\bar{\mathbf{w}}_i^P$.

**Lemma 2** Given the definition of $\mathbf{w}_i^P$ and $\bar{\mathbf{w}}_i^P$ in (5) and (7), let $\mathcal{X}$ be a compact set with radius $R$, i.e., $\forall \mathbf{x}, \mathbf{x}' \in \mathcal{X}, \|\mathbf{x} - \mathbf{x}'\| \le R$. For any $\delta \in (0, 1]$ with probability $\ge 1 - \delta$, we have

$$\|\mathbf{w}_i^P - \bar{\mathbf{w}}_i^P\|^2 \le \sum_{k=1}^K \left( \frac{\alpha R}{\sqrt{m_{ik}}}(1 + \sqrt{\frac{1}{2}\log(\frac{1}{\delta})}) + O_{\alpha,\beta}(\gamma, C) \right)^2. \tag{24}$$

**Proof** According to the definition of $\mathbf{w}^P, \bar{\mathbf{w}}^P$ and $\hat{\mathbf{w}}^P$, we have

$$\|\mathbf{w}_i^P - \bar{\mathbf{w}}_i^P\|^2$$

$$= \sum_{k=1}^K \|\mathbf{w}_i^P[k] - \bar{\mathbf{w}}_i^P[k]\|^2$$

$$= \sum_{k=1}^K \|\mathbb{E}[\hat{\Phi}_{\mathbf{v}}(S_{ik})] - \hat{\Phi}_{\mathbf{v}}(S_{ik}) + \hat{\Phi}_{\mathbf{v}}(S_{ik}) - \bar{\Phi}_{\mathbf{v}}(S_{ik})\|^2$$

$$= \sum_{k=1}^K \left( \|\mathbb{E}[\hat{\Phi}_{\mathbf{v}}(S_{ik})] - \hat{\Phi}_{\mathbf{v}}(S_{ik})\|^2 + \|\hat{\Phi}_{\mathbf{v}}(S_{ik}) - \bar{\Phi}_{\mathbf{v}}(S_{ik})\|^2 + 2(\mathbb{E}[\hat{\Phi}_{\mathbf{v}}(S_{ik})] - \hat{\Phi}_{\mathbf{v}}(S_{ik}))^\top (\hat{\Phi}_{\mathbf{v}}(S_{ik}) - \bar{\Phi}_{\mathbf{v}}(S_{ik})) \right)$$

$$\le \sum_{k=1}^K \left( \|\mathbb{E}[\hat{\Phi}_{\mathbf{v}}(S_{ik})] - \hat{\Phi}_{\mathbf{v}}(S_{ik})\|^2 + \|\hat{\Phi}_{\mathbf{v}}(S_{ik}) - \bar{\Phi}_{\mathbf{v}}(S_{ik})\|^2 + 2\|\mathbb{E}[\hat{\Phi}_{\mathbf{v}}(S_{ik})] - \hat{\Phi}_{\mathbf{v}}(S_{ik})\|\|\hat{\Phi}_{\mathbf{v}}(S_{ik}) - \bar{\Phi}_{\mathbf{v}}(S_{ik})\| \right).$$

$$\tag{25}$$

Substitute Lemma 3 and Lemma 1 into the above inequality, we can derive

$$\mathbb{P}_{S_{ik} \sim D_k^{m_k}} \left\{ \|\mathbf{w}^P - \bar{\mathbf{w}}^P\|^2 \le \sum_{k=1}^K \left( \frac{\alpha R}{\sqrt{m_{ik}}}(1 + \sqrt{\frac{1}{2}\log(\frac{1}{\delta})}) + O_{\alpha,\beta}(\gamma, C) \right)^2 \right\} \ge 1 - \delta. \tag{26}$$

432 This gives the assertion.

433 Lemma 2 shows that the approximation error between expected prior predictor and LCC-based prior
434 predictor depends on the number of samples in each category and the quality of the LCC coding
435 scheme.

### B.3 PROOF OF THEOREM 3

**Theorem 3** Let $Q$ be the posterior of base learner $Q = \mathcal{N}(\mathbf{w}^Q, \sigma_{\mathbf{w}}^2 I_{d_{\mathbf{w}}})$ and $P$ be the prior $\mathcal{N}(\bar{\Phi}_{\mathbf{v}}(S), \sigma_{\mathbf{w}}^2 I_{d_{\mathbf{w}}})$. The mean of prior is produced by the LCC-based prior predictor $\bar{\Phi}_{\mathbf{v}}(S)$ in Eq. (7) and its parameter $\mathbf{v}$ is sampled from the hyperposterior of meta learner $\mathcal{Q} = \mathcal{N}(\mathbf{v}^{\mathcal{Q}}, \sigma_{\mathbf{v}}^2 I_{d_{\mathbf{v}}})$. Give the hyperprior $\mathcal{P} = \mathcal{N}(0, \sigma_{\mathbf{v}}^2 I_{d_{\mathbf{v}}})$, then for any hyperposterior $\mathcal{Q}$, any $c_1, c_2 > 0$ and any $\delta \in (0, 1]$ with probability $\geq 1 - \delta$ we have,

$$
er(\mathcal{Q}) \leq c_1' c_2' \hat{er}(\mathcal{Q}) + (\sum_{i=1}^n \frac{c_1' c_2'}{2c_2 nm_i \sigma_{\mathbf{v}}^2} + \frac{c_1'}{2c_1 n\sigma_{\mathbf{v}}^2})\|\mathbf{v}^{\mathcal{Q}}\|^2 + \sum_{i=1}^n \frac{c_1' c_2'}{c_2 nm_i \sigma_{\mathbf{w}}^2}\|\mathbb{E}_{\mathbf{v}}\mathbf{w}_i^Q - \bar{\Phi}_{\mathbf{v}^{\mathcal{Q}}}(S_i)\|^2
$$

$$
+ \sum_{i=1}^n \frac{c_1' c_2'}{c_2 nm_i \sigma_{\mathbf{w}}^2} \left( \frac{1}{\sigma_{\mathbf{w}}^2} \sum_{k=1}^K \left( \frac{\alpha R}{\sqrt{m_{ik}}}(1 + \sqrt{\frac{1}{2}\log(\frac{4n}{\delta})}) + O_{\alpha,\beta}(\gamma, C) \right)^2 + d_{\mathbf{w}} K(\frac{\sigma_{\mathbf{v}}}{\sigma_{\mathbf{w}}})^2 \right)
$$

$$
+ \sum_{i=1}^n \frac{c_1' c_2'}{c_2 nm_i \sigma_{\mathbf{w}}^2} \log \frac{4n}{\delta} + \frac{c_1'}{2c_1 n\sigma_{\mathbf{v}}^2} \log \frac{2}{\delta}, \tag{27}
$$

where $c_1' = \frac{c_1}{1-e^{-c_1}}$ and $c_2' = \frac{c_2}{1-e^{-c_2}}$. We can simplify the notation and obtain that

$$
er(\mathcal{Q}) \leq c_1' c_2' \hat{er}(\mathcal{Q}) + (\sum_{i=1}^n \frac{c_1' c_2'}{2c_2 nm_i \sigma_{\mathbf{v}}^2} + \frac{c_1'}{2c_1 n\sigma_{\mathbf{v}}^2})\|\mathbf{v}^{\mathcal{Q}}\|^2 + \sum_{i=1}^n \frac{c_1' c_2'}{c_2 nm_i \sigma_{\mathbf{w}}^2}\|\mathbb{E}_{\mathbf{v}}\mathbf{w}_i^Q - \bar{\Phi}_{\mathbf{v}^{\mathcal{Q}}}(S_i)\|^2
$$

$$
+ const(\alpha, \beta, R, \delta, n, m_i). \tag{28}
$$

437 **Proof** Our proof contains two steps. First, we bound the error within observed tasks due to ob-
438 serving a limited number of samples. Then we bound the error on the task environment level due
439 to observing a finite number of tasks. Both of the two steps utilize Catoni's classical PAC-Bayes
440 bound (Catoni, 2007) to measure the error. We give here a general statement of the Catoni's classical
441 PAC-Bayes bound.

**Theorem 5.** *(Classical PAC-Bayes bound, general notations) Let $\mathcal{X}$ be a sample space and $\mathbb{X}$ be some distribution over $\mathcal{X}$, and let $\mathcal{F}$ be a hypotheses space of functions over $\mathcal{X}$. Define a loss function $g(f, X) : \mathcal{F} \times \mathcal{X} \to [0, 1]$, and let $X_1^G \triangleq \{X_1, \dots, X_G\}$ be a sequence of $K$ independent random variables distributed according to $\mathbb{X}$. Let $\pi$ be some prior distribution over $\mathcal{F}$ (which must not depend on the samples $X_1, \dots, X_k$). For any $\delta \in (0, 1]$, the following bounds holds uniformly for all posterior distribution $\rho$ over $\mathcal{F}$ (even sample dependent),*

$$
\mathbb{P}_{X_1^K \underset{i.i.d}{\sim} \mathbb{X}} \left\{ \mathbb{E}_{X \sim \mathbb{X}} \mathbb{E}_{f \sim \rho} g(f, X) \leq \frac{c}{1-e^{-c}} \left[ \frac{1}{G} \sum_{g=1}^G \mathbb{E}_{f \sim \rho} g(f, X_k) + \frac{KL(\rho||\pi) + \log\frac{1}{\delta}}{K \times c} \right], \forall \rho \right\}
$$

$$
\geq 1 - \delta. \tag{29}
$$

442 **First step** We utilize Theorem 5 to bound the generalization error in each of the observed tasks.
443 Let $i \in 1, \dots, n$ be the index of task. For task i, we substitute the following definition into
444 the Catoni's PAC-Bayes Bound. Specifically, $X_g \triangleq (\mathbf{x}_{ij}, y_{ij}), K \triangleq m_i$ denote the samples and
445 $\mathbb{X} \triangleq D_i$ denotes the data distribution. We instantiate the hypotheses with a hierarchical model
446 $f \triangleq (\mathbf{v}, \mathbf{w})$, where $\mathbf{v} \in \mathbb{R}^{d_{\mathbf{v}}}$ and $\mathbf{w} \in \mathbb{R}^{d_{\mathbf{w}}}$ are the parameters of meta learner (prior predic-
447 tor) $\Phi_{\mathbf{v}}(\cdot)$ and base learner $h(\cdot)$ respectively. The loss function only considers the base learner,
448 which is defined as $g(f, X) \triangleq \ell(h_{\mathbf{w}}(\mathbf{x}), y)$. The prior over model parameter is represented
449 as $\pi \triangleq (\mathcal{P}, P) \triangleq (\mathcal{N}(0, \sigma_{\mathbf{v}}^2 I_{d_{\mathbf{v}}}), \mathcal{N}(\mathbf{w}^P, \sigma_{\mathbf{w}}^2 I_{d_{\mathbf{w}}}))$, a Gaussian distribution (hyperprior of meta

learner) centered at $0$ and a Gaussian distribution (prior of base learner) centered at $\mathbf{w}^P$, respectively. We set the posterior to $\rho \triangleq (\mathcal{Q}, Q) \triangleq (\mathcal{N}(\mathbf{v}^{\mathcal{Q}}, \sigma_{\mathbf{v}}^2 I_{d_{\mathbf{v}}}), \mathcal{N}(\mathbf{w}^Q, \sigma_{\mathbf{w}}^2 I_{d_{\mathbf{w}}}))$, a Gaussian distribution (hyperposterior of meta learner) centered at $\mathbf{v}^{\mathcal{Q}}$ and a Gaussian distribution (posterior of base learner) centered at $\mathbf{w}^Q$. According to Theorem 5, the generalization bound holds for any posterior distribution including the one generated in our localized meta-learning framework. Specifically, we first sample $\mathbf{v}$ from hyperposterior $\mathcal{N}(\mathbf{v}^{\mathcal{Q}}, \sigma_{\mathbf{v}}^2 I_{d_{\mathbf{v}}})$ and estimate $\mathbf{w}^P$ by leveraging expected prior predictor $\mathbf{w}^P = \Phi_{\mathbf{v}}(D)$. The base learner algorithm $A_b(S, P)$ utilizes the sample set $S$ and the prior $P = \mathcal{N}(\mathbf{w}^P, \sigma_{\mathbf{w}}^2 I_{d_{\mathbf{w}}})$ to produce a posterior $Q = A_b(S, P) = \mathcal{N}(\mathbf{w}^Q, \sigma_{\mathbf{w}}^2 I_{d_{\mathbf{w}}})$. Then we sample base learner parameter $\mathbf{w}$ from posterior $\mathcal{N}(\mathbf{w}^Q, \sigma_{\mathbf{w}}^2 I_{d_{\mathbf{w}}})$ and compute the incurred loss $\ell(h_{\mathbf{w}}(\mathbf{x}), y)$. On the whole, meta-learning algorithm $A_m(S_1, \ldots, S_n, \mathcal{P})$ observes a series of tasks $S_1, \ldots, S_n$ and adjusts its hyperprior $\mathcal{P} = \mathcal{N}(\mathbf{v}^{\mathcal{P}}, \sigma_{\mathbf{v}}^2 I_{d_{\mathbf{v}}})$ into hyperposterior $\mathcal{Q} = A_m(S_1, \ldots, S_n, \mathcal{P}) = \mathcal{N}(\mathbf{v}^{\mathcal{Q}}, \sigma_{\mathbf{v}}^2 I_{d_{\mathbf{v}}})$.

The KL divergence term between prior $\pi$ and posterior $\rho$ is computed as follows:

$$
\begin{aligned}
KL(\rho \| \pi) &= \mathop{\mathbb{E}}_{f \sim \rho} \log \frac{\rho(f)}{\pi(f)} = \mathop{\mathbb{E}}_{\mathbf{v} \sim \mathcal{N}(\mathbf{v}^{\mathcal{Q}}, \sigma_{\mathbf{v}}^2 I_{d_{\mathbf{v}}})} \mathop{\mathbb{E}}_{\mathbf{w} \sim \mathcal{N}(\mathbf{w}^Q, \sigma_{\mathbf{w}}^2 I_{d_{\mathbf{w}}})} \log \frac{\mathcal{N}(\mathbf{v}^{\mathcal{Q}}, \sigma_{\mathbf{v}}^2 I_{d_{\mathbf{v}}}) \mathcal{N}(\mathbf{w}^Q, \sigma_{\mathbf{w}}^2 I_{d_{\mathbf{w}}})}{\mathcal{N}(0, \sigma_{\mathbf{v}}^2 I_{d_{\mathbf{v}}}) \mathcal{N}(\mathbf{w}^P, \sigma_{\mathbf{w}}^2 I_{d_{\mathbf{w}}})} \\
&= \mathop{\mathbb{E}}_{\mathbf{v} \sim \mathcal{N}(\mathbf{v}^{\mathcal{Q}}, \sigma_{\mathbf{v}}^2 I_{d_{\mathbf{v}}})} \log \frac{\mathcal{N}(\mathbf{v}^{\mathcal{Q}}, \sigma_{\mathbf{v}}^2 I_{d_{\mathbf{v}}})}{\mathcal{N}(0, \sigma_{\mathbf{v}}^2 I_{d_{\mathbf{v}}})} + \mathop{\mathbb{E}}_{\mathbf{v} \sim \mathcal{N}(\mathbf{v}^{\mathcal{Q}}, \sigma_{\mathbf{v}}^2 I_{d_{\mathbf{v}}})} \mathop{\mathbb{E}}_{\mathbf{w} \sim \mathcal{N}(\mathbf{w}^Q, \sigma_{\mathbf{w}}^2 I_{d_{\mathbf{w}}})} \log \frac{\mathcal{N}(\mathbf{w}^Q, \sigma_{\mathbf{w}}^2 I_{d_{\mathbf{w}}})}{\mathcal{N}(\mathbf{w}^P, \sigma_{\mathbf{w}}^2 I_{d_{\mathbf{w}}})} \\
&= \frac{1}{2\sigma_{\mathbf{v}}^2} \|\mathbf{v}^{\mathcal{Q}}\|^2 + \mathop{\mathbb{E}}_{\mathbf{v} \sim \mathcal{N}(\mathbf{v}^{\mathcal{Q}}, \sigma_{\mathbf{v}}^2 I_{d_{\mathbf{v}}})} \frac{1}{2\sigma_{\mathbf{w}}^2} \|\mathbf{w}^Q - \mathbf{w}^P\|^2.
\end{aligned} \tag{30}
$$

In our localized meta-learning framework, in order to make $KL(Q \| P)$ small, the center of prior distribution $\mathbf{w}^P$ is generated by the expected prior predictor $\mathbf{w}^P = \Phi_{\mathbf{v}}(D)$. However, the data distribution $D$ is considered unknown and our only insight as to $D_{ik}$ is through the sample set $S_{ik}$. In this work, we approximate the expected prior predictor $\Phi_{\mathbf{v}}(D)$ with the LCC-based prior predictor $\bar{\mathbf{w}}^P = \bar{\Phi}_{\mathbf{v}}(S)$. Denote the term $\mathop{\mathbb{E}}_{\mathbf{v} \sim \mathcal{N}(\mathbf{v}^{\mathcal{Q}}, \sigma_{\mathbf{v}}^2 I_{d_{\mathbf{v}}})} \frac{1}{2\sigma_{\mathbf{w}}^2} \|\mathbf{w}^Q - \mathbf{w}^P\|^2$ by $\mathop{\mathbb{E}}_{\mathbf{v}} \frac{1}{2\sigma_{\mathbf{w}}^2} \|\mathbf{w}^Q - \mathbf{w}^P\|^2$ for convenience, we have

$$
\begin{aligned}
\mathop{\mathbb{E}}_{\mathbf{v}} \frac{1}{2\sigma_{\mathbf{w}}^2} \|\mathbf{w}^Q - \mathbf{w}^P\|^2 &= \mathop{\mathbb{E}}_{\mathbf{v}} \frac{1}{2\sigma_{\mathbf{w}}^2} \|\mathbf{w}^Q - \bar{\mathbf{w}}^P + \bar{\mathbf{w}}^P - \mathbf{w}^P\|^2 \\
&= \mathop{\mathbb{E}}_{\mathbf{v}} \frac{1}{2\sigma_{\mathbf{w}}^2} [\|\mathbf{w}^Q - \bar{\mathbf{w}}^P\|^2 + \|\bar{\mathbf{w}}^P - \mathbf{w}^P\|^2 + 2(\mathbf{w}^Q - \bar{\mathbf{w}}^P)^\top (\bar{\mathbf{w}}^P - \mathbf{w}^P)] \\
&\leq \mathop{\mathbb{E}}_{\mathbf{v}} \frac{1}{2\sigma_{\mathbf{w}}^2} [\|\mathbf{w}^Q - \bar{\mathbf{w}}^P\|^2 + \|\bar{\mathbf{w}}^P - \mathbf{w}^P\|^2 + 2\|\mathbf{w}^Q - \bar{\mathbf{w}}^P\| \|\bar{\mathbf{w}}^P - \mathbf{w}^P\|] \\
&\leq \frac{1}{\sigma_{\mathbf{w}}^2} \mathop{\mathbb{E}}_{\mathbf{v}} \|\mathbf{w}^Q - \bar{\Phi}_{\mathbf{v}}(S)\|^2 + \frac{1}{\sigma_{\mathbf{w}}^2} \mathop{\mathbb{E}}_{\mathbf{v}} \|\bar{\mathbf{w}}^P - \mathbf{w}^P\|^2.
\end{aligned} \tag{31}
$$

Since $\bar{\mathbf{w}}_i^P = \bar{\Phi}_{\mathbf{v}}(S_i) = [\bar{\Phi}_{\mathbf{v}}(S_{i1}), \ldots, \bar{\Phi}_{\mathbf{v}}(S_{ik}), \ldots, \bar{\Phi}_{\mathbf{v}}(S_{iK})]$, we have

$$
\begin{aligned}
\mathop{\mathbb{E}}_{\mathbf{v}} \|\mathbf{w}_i^Q - \bar{\Phi}_{\mathbf{v}}(S_i)\|^2 &= \sum_{k=1}^{K} \mathop{\mathbb{E}}_{\mathbf{v}} \|\mathbf{w}_i^Q[k] - \bar{\Phi}_{\mathbf{v}}(S_{ik})\|^2 \\
&= \sum_{k=1}^{K} \left( \mathop{\mathbb{E}}_{\mathbf{v}} \|\mathbf{w}_i^Q[k]\|^2 - 2(\mathop{\mathbb{E}}_{\mathbf{v}} \mathbf{w}_i^Q[k])^\top (\bar{\Phi}_{\mathbf{v}^{\mathcal{Q}}}(S_{ik})) + \|\bar{\Phi}_{\mathbf{v}^{\mathcal{Q}}}(S_{ik})\|^2 + \mathop{\mathbb{V}}_{\mathbf{v}}[\|\bar{\Phi}_{\mathbf{v}}(S_{ik})\|] \right) \\
&= \sum_{k=1}^{K} \left( \|\mathop{\mathbb{E}}_{\mathbf{v}} \mathbf{w}_i^Q[k] - \bar{\Phi}_{\mathbf{v}^{\mathcal{Q}}}(S_{ik})\|^2 + \frac{d_{\mathbf{v}}}{|C|} \sigma_{\mathbf{v}}^2 \right) \\
&= \|\mathop{\mathbb{E}}_{\mathbf{v}} \mathbf{w}_i^Q - \bar{\Phi}_{\mathbf{v}^{\mathcal{Q}}}(S_i)\|^2 + d_{\mathbf{w}} K \sigma_{\mathbf{v}}^2,
\end{aligned} \tag{32}
$$

where $\underset{\mathbf{v}}{\mathbb{V}}[\|\bar{\Phi}_{\mathbf{v}}(S_{ik})\|]$ denotes the variance of $\|\bar{\Phi}_{\mathbf{v}}(S_{ik})\|$. The last equality uses the fact that $d_{\mathbf{v}} = |C|d_{\mathbf{w}}$. Combining Lemma 2, for any $\delta' \in (0,1]$ with probability $\geq 1 - \delta'$ we have

$$\underset{\mathbf{v}}{\mathbb{E}}\frac{1}{2\sigma_{\mathbf{w}}^2}\|\mathbf{w}_i^Q - \mathbf{w}_i^P\|^2$$

$$\leq \frac{1}{\sigma_{\mathbf{w}}^2}\|\underset{\mathbf{v}}{\mathbb{E}}\mathbf{w}_i^Q - \bar{\Phi}_{\mathbf{v}^Q}(S_i)\|^2 + d_{\mathbf{w}}K(\frac{\sigma_{\mathbf{v}}}{\sigma_{\mathbf{w}}})^2 + \frac{1}{\sigma_{\mathbf{w}}^2}\sum_{k=1}^{K}\left(\frac{\alpha R}{\sqrt{m_{ik}}}(1 + \sqrt{\frac{1}{2}\log(\frac{1}{\delta})}) + O_{\alpha,\beta}(\gamma, C)\right)^2$$

$$\tag{33}$$

Then, according to Theorem 5, we obtain that for any $\frac{\delta_i}{2} > 0$

$$\mathbb{P}_{S_i \sim D_i^{m_i}}\left\{\underset{(\mathbf{x},y)\sim D_i}{\mathbb{E}}\underset{\mathbf{v}\sim\mathcal{N}(\mathbf{v}^Q,\sigma_{\mathbf{v}}^2 I_{d_{\mathbf{v}}})}{\mathbb{E}}\underset{\mathbf{w}\sim\mathcal{N}(\mathbf{w}^Q,\sigma_{\mathbf{w}}^2 I_{d_{\mathbf{w}}})}{\mathbb{E}}\ell(h_{\mathbf{w}}(\mathbf{x}),y)\right.$$

$$\leq \frac{c_2}{1-e^{-c_2}}\cdot\frac{1}{m_i}\sum_{j=1}^{m_i}\underset{\mathbf{v}\sim\mathcal{N}(\mathbf{v}^Q,\sigma_{\mathbf{v}}^2 I_{d_{\mathbf{v}}})}{\mathbb{E}}\underset{\mathbf{w}\sim\mathcal{N}(\mathbf{w}^Q,\sigma_{\mathbf{w}}^2 I_{d_{\mathbf{w}}})}{\mathbb{E}}\ell(h_{\mathbf{w}}(\mathbf{x}_j),y_j)$$

$$\left.+ \frac{1}{(1-e^{-c_2})\cdot m_i}\left(\frac{1}{2\sigma_{\mathbf{v}}^2}\|\mathbf{v}^Q\|^2 + \underset{\mathbf{v}\sim\mathcal{N}(\mathbf{v}^Q,\sigma_{\mathbf{v}}^2 I_{d_{\mathbf{v}}})}{\mathbb{E}}\frac{1}{2\sigma_{\mathbf{w}}^2}\|\mathbf{w}_i^Q - \mathbf{w}_i^P\|^2 + \log\frac{2}{\delta_i}\right), \forall\mathcal{Q}\right\} \geq 1 - \frac{\delta_i}{2},$$

$$\tag{34}$$

for all observed tasks $i = 1, \ldots, n$. Define $\delta' = \frac{\delta_i}{2}$ and combine inequality (33), we obtain

$$\mathbb{P}_{S_i \sim D_i^{m_i}}\left\{\underset{(\mathbf{x},y)\sim D_i}{\mathbb{E}}\underset{\mathbf{v}\sim\mathcal{N}(\mathbf{v}^Q,\sigma_{\mathbf{v}}^2 I_{d_{\mathbf{v}}})}{\mathbb{E}}\underset{\mathbf{w}\sim\mathcal{N}(\mathbf{w}^Q,\sigma_{\mathbf{w}}^2 I_{d_{\mathbf{w}}})}{\mathbb{E}}\ell(h_{\mathbf{w}}(\mathbf{x}),y)\right.$$

$$\leq \frac{c_2}{1-e^{-c_2}}\cdot\frac{1}{m_i}\sum_{j=1}^{m_i}\underset{\mathbf{v}\sim\mathcal{N}(\mathbf{v}^Q,\sigma_{\mathbf{v}}^2 I_{d_{\mathbf{v}}})}{\mathbb{E}}\underset{\mathbf{w}\sim\mathcal{N}(\mathbf{w}^Q,\sigma_{\mathbf{w}}^2 I_{d_{\mathbf{w}}})}{\mathbb{E}}\ell(h_{\mathbf{w}}(\mathbf{x}_j),y_j)$$

$$+ \frac{1}{(1-e^{-c_2})m_i}\cdot\left(\frac{1}{2\sigma_{\mathbf{v}}^2}\|\mathbf{v}^Q\|^2 + \frac{1}{\sigma_{\mathbf{w}}^2}\|\underset{\mathbf{v}}{\mathbb{E}}\mathbf{w}_i^Q - \bar{\Phi}_{\mathbf{v}^Q}(S_i)\|^2 + \log\frac{2}{\delta_i} + d_{\mathbf{w}}K(\frac{\sigma_{\mathbf{v}}}{\sigma_{\mathbf{w}}})^2\right.$$

$$\left.\left.+ \frac{1}{\sigma_{\mathbf{w}}^2}\sum_{k=1}^{K}\left(\frac{\alpha R}{\sqrt{m_{ik}}}(1 + \sqrt{\frac{1}{2}\log(\frac{2}{\delta_i})}) + O_{\alpha,\beta}(\gamma, C)\right)^2\right), \forall\mathcal{Q}\right\} \geq 1 - \delta_i,\tag{35}$$

Using the notations in Section 4, the above bound can be simplified as

$$\mathbb{P}_{S_i \sim D_i^{m_i}}\left\{\underset{\mathbf{v}\sim\mathcal{N}(\mathbf{v}^Q,\sigma_{\mathbf{v}}^2 I_{d_{\mathbf{v}}}),\mathbf{w}^P=\Phi_{\mathbf{v}}(D),P_i=\mathcal{N}(\mathbf{w}^P,\sigma_{\mathbf{w}}^2 I_{d_{\mathbf{w}}})}{\mathbb{E}}er(A_b(S_i,P_i))\right.$$

$$\leq \frac{c_2}{1-e^{-c_2}}\underset{\mathbf{v}\sim\mathcal{N}(\mathbf{v}^Q,\sigma_{\mathbf{v}}^2 I_{d_{\mathbf{v}}}),\mathbf{w}^P=\Phi_{\mathbf{v}}(D),P_i=\mathcal{N}(\mathbf{w}^P,\sigma_{\mathbf{w}}^2 I_{d_{\mathbf{w}}})}{\mathbb{E}}\hat{er}(A_b(S_i,P_i))$$

$$+ \frac{1}{(1-e^{-c_2})m_i}\left(\frac{1}{2\sigma_{\mathbf{v}}^2}\|\mathbf{v}^Q\|^2 + \frac{1}{\sigma_{\mathbf{w}}^2}\|\underset{\mathbf{v}}{\mathbb{E}}\mathbf{w}_i^Q - \bar{\Phi}_{\mathbf{v}^Q}(S_i)\|^2 + \log\frac{2}{\delta_i} + d_{\mathbf{w}}K(\frac{\sigma_{\mathbf{v}}}{\sigma_{\mathbf{w}}})^2\right.$$

$$\left.\left.+ \frac{1}{\sigma_{\mathbf{w}}^2}\sum_{k=1}^{K}\left(\frac{\alpha R}{\sqrt{m_{ik}}}(1 + \sqrt{\frac{1}{2}\log(\frac{2}{\delta_i})}) + O_{\alpha,\beta}(\gamma, C)\right)^2\right), \forall\mathcal{Q}\right\} \geq 1 - \delta_i.\tag{36}$$

**Second step** Next we bound the error due to observing a limited number of tasks from the environment. We reuse Theorem 5 with the following substitutions. The samples are $(D_i, m_i, S_i)$, $i = 1, \ldots, n$, where $(D_i, m_i)$ are sampled from the same meta distribution $\tau$ and $S_i \sim D_i^{m_i}$. The hyposthesis is parameterized as $\Phi_{\mathbf{v}}(D)$ with meta learner parameter $\mathbf{v}$. The loss function is $g(f,X) \triangleq \underset{(\mathbf{x},y)\sim D}{\mathbb{E}}\underset{\mathbf{w}\sim\mathcal{N}(\mathbf{w}^Q,\sigma_{\mathbf{w}}^2 I_{d_{\mathbf{w}}})}{\mathbb{E}}\ell(h_{\mathbf{w}}(\mathbf{x}),y)$, where $\mathbf{w}^Q = A_b(S_i,P_i)$. Let $\pi \triangleq \mathcal{N}(0,\sigma_{\mathbf{v}}^2 I_{d_{\mathbf{v}}})$ be

the prior over meta learner parameter, the following holds for any $\delta_0 > 0$,

$$\mathbb{P}_{(D_i^{m_i}) \sim \tau, S_i \sim D_i^{m_i}, i=1,\dots,n} \left\{ \mathbb{E}_{(D,m) \sim \tau} \mathbb{E}_{S \sim D^m} \mathbb{E}_{\mathbf{v} \sim \mathcal{N}(\mathbf{v}^{\mathcal{Q}}, \sigma_{\mathbf{v}}^2 I_{d_{\mathbf{v}}})} \mathbb{E}_{\mathbf{w} \sim \mathcal{N}(\mathbf{w}^Q, \sigma_{\mathbf{w}}^2 I_{d_{\mathbf{w}}})} \mathbb{E}_{(x,y) \sim D_i} \ell(h_{\mathbf{w}}(\mathbf{x}), y) \right.$$

$$\leq \frac{c_1}{1 - e^{-c_1}} \cdot \frac{1}{n} \sum_{i=1}^{n} \mathbb{E}_{\mathbf{v} \sim \mathcal{N}(\mathbf{v}^{\mathcal{Q}}, \sigma_{\mathbf{v}}^2 I_{d_{\mathbf{v}}})} \mathbb{E}_{\mathbf{w} \sim \mathcal{N}(\mathbf{w}^Q, \sigma_{\mathbf{w}}^2 I_{d_{\mathbf{w}}})} \mathbb{E}_{(x,y) \sim D_i} \ell(h_{\mathbf{w}}(\mathbf{x}), y)$$

$$+ \frac{1}{(1 - e^{-c_1})n} \left( \frac{1}{2\sigma_{\mathbf{v}}^2} \|\mathbf{v}^{\mathcal{Q}}\|^2 + \log \frac{1}{\delta_0} \right), \forall \mathcal{Q} \left. \vphantom{\sum} \right\} \geq 1 - \delta_0. \tag{37}$$

Using the term in Section 4, the above bound can be simplified as

$$\mathbb{P}_{(D_i^{m_i}) \sim \tau, S_i \sim D_i^{m_i}, i=1,\dots,n} \left\{ er(\mathcal{Q}) \right.$$

$$\leq \frac{c_1}{1 - e^{-c_1}} \cdot \frac{1}{n} \sum_{i=1}^{n} \mathbb{E}_{\mathbf{v} \sim \mathcal{N}(\mathbf{v}^{\mathcal{Q}}, \sigma_{\mathbf{v}}^2 I_{d_{\mathbf{v}}}), \mathbf{w}^P = \Phi_{\mathbf{v}}(D), P_i = \mathcal{N}(\mathbf{w}^P, \sigma_{\mathbf{w}}^2 I_{d_{\mathbf{w}}})} er(A_b(S_i, P_i))$$

$$+ \frac{1}{(1 - e^{-c_1})n} \left( \frac{1}{2\sigma_{\mathbf{v}}^2} \|\mathbf{v}^{\mathcal{Q}}\|^2 + \log \frac{1}{\delta_0} \right), \forall \mathcal{Q} \left. \vphantom{\sum} \right\} \geq 1 - \delta_0, \tag{38}$$

Finally, by employing the union bound, we could bound the probability of the intersection of the events in (36) and (38) For any $\delta > 0$, set $\delta_0 \triangleq \frac{\delta}{2}$ and $\delta_i \triangleq \frac{\delta}{2n}$ for $i = 1, \dots, n$, we have

$$\mathbb{P}_{(D_i^{m_i}) \sim \tau, S_i \sim D_i^{m_i}, i=1,\dots,n} \left\{ er(\mathcal{Q}) \right.$$

$$\leq \frac{c_1 c_2}{(1 - e^{-c_1})(1 - e^{-c_2})} \cdot \frac{1}{n} \sum_{i=1}^{n} \mathbb{E}_{\mathbf{v} \sim \mathcal{N}(\mathbf{v}^{\mathcal{Q}}, \sigma_{\mathbf{v}}^2 I_{d_{\mathbf{v}}}), \mathbf{w}^P = \Phi_{\mathbf{v}}(D), P_i = \mathcal{N}(\mathbf{w}^P, \sigma_{\mathbf{w}}^2 I_{d_{\mathbf{w}}})} \hat{er}(A_b(S_i, P_i))$$

$$+ \frac{c_1}{1 - e^{-c_1}} \cdot \frac{1}{n} \sum_{i=1}^{n} \frac{1}{(1 - e^{-c_2})m_i} \left( \frac{1}{2\sigma_{\mathbf{v}}^2} \|\mathbf{v}^{\mathcal{Q}}\|^2 + \frac{1}{\sigma_{\mathbf{w}}^2} \|\mathbb{E}_{\mathbf{v}} \mathbf{w}_i^Q - \bar{\Phi}_{\mathbf{v}^{\mathcal{Q}}}(S_i)\|^2 + \log \frac{4n}{\delta} \right.$$

$$+ \frac{1}{\sigma_{\mathbf{w}}^2} \sum_{k=1}^{K} \left( \frac{\alpha R}{\sqrt{m_{ik}}} (1 + \sqrt{\frac{1}{2} \log(\frac{4n}{\delta})}) + O_{\alpha,\beta}(\gamma, C) \right)^2 + d_{\mathbf{w}} K (\frac{\sigma_{\mathbf{v}}}{\sigma_{\mathbf{w}}})^2 \right)$$

$$+ \frac{1}{(1 - e^{-c_1})n} \left( \frac{1}{2\sigma_{\mathbf{v}}^2} \|\mathbf{v}^{\mathcal{Q}}\|^2 + \log \frac{2}{\delta} \right), \forall \mathcal{Q} \left. \vphantom{\sum} \right\} \geq 1 - \delta. \tag{39}$$

We can further simplify the notation and obtain that

$$\mathbb{P}_{(D_i^{m_i}) \sim \tau, S_i \sim D_i^{m_i}, i=1,\dots,n} \left\{ er(\mathcal{Q}) \leq c_1' c_2' \hat{er}(\mathcal{Q}) \right.$$

$$+ (\sum_{i=1}^{n} \frac{c_1' c_2'}{2c_2 n m_i \sigma_{\mathbf{v}}^2} + \frac{c_1'}{2c_1 n \sigma_{\mathbf{v}}^2}) \|\mathbf{v}^{\mathcal{Q}}\|^2 + \sum_{i=1}^{n} \frac{c_1' c_2'}{c_2 n m_i \sigma_{\mathbf{w}}^2} \|\mathbb{E}_{\mathbf{v}} \mathbf{w}_i^Q - \bar{\Phi}_{\mathbf{v}^{\mathcal{Q}}}(S_i)\|^2$$

$$+ const(\alpha, \beta, R, \delta, n, m_i), \forall \mathcal{Q} \left. \vphantom{\sum} \right\} \geq 1 - \delta, \tag{40}$$

where $c_1' = \frac{c_1}{1 - e^{-c_1}}$ and $c_2' = \frac{c_2}{1 - e^{-c_2}}$. This completes the proof.

### B.4 PROOF OF THEOREM 2

**Theorem 2** Let $Q$ be the posterior of base learner $Q = \mathcal{N}(\mathbf{w}^Q, \sigma_{\mathbf{w}}^2 I_{d_{\mathbf{w}}})$ and $P$ be the prior $\mathcal{N}(\mathbf{w}^P, \sigma_{\mathbf{w}}^2 I_{d_{\mathbf{w}}})$. The mean of prior is sampled from the hyperposterior of meta learner $\mathcal{Q} = \mathcal{N}(\mathbf{w}^{\mathcal{Q}}, \sigma_{\mathbf{w}}^2 I_{d_{\mathbf{w}}})$. Give the hyperprior $\mathcal{P} = \mathcal{N}(0, \sigma_{\mathbf{w}}^2 I_{d_{\mathbf{w}}})$, then for any hyperposterior $\mathcal{Q}$, any

$c_1, c_2 > 0$ and any $\delta \in (0, 1]$ with probability $\geq 1 - \delta$ we have,

$$
\begin{aligned}
er(\mathcal{Q}) \leq & c_1' c_2' \hat{er}(\mathcal{Q}) + \left( \sum_{i=1}^{n} \frac{c_1' c_2'}{2 c_2 n m_i \sigma_{\mathbf{w}}^2} + \frac{c_1'}{2 c_1 n \sigma_{\mathbf{w}}^2} \right) \|\mathbf{w}^{\mathcal{Q}}\|^2 + \sum_{i=1}^{n} \frac{c_1' c_2'}{2 c_2 n m_i \sigma_{\mathbf{w}}^2} \| \mathop{\mathbb{E}}_{\mathbf{w}^P} \mathbf{w}_i^Q - \mathbf{w}^{\mathcal{Q}} \|^2 \\
& + \sum_{i=1}^{n} \frac{c_1' c_2'}{c_2 n m_i \sigma_{\mathbf{w}}^2} \left( \frac{1}{2} + \log \frac{2n}{\delta} \right) + \frac{c_1'}{c_1 n \sigma_{\mathbf{w}}^2} \log \frac{2}{\delta},
\end{aligned} \tag{41}
$$

where $c_1' = \frac{c_1}{1 - e^{-c_1}}$ and $c_2' = \frac{c_2}{1 - e^{-c_2}}$.

**Proof** Instead of generating the mean of prior with a prior predictor, the vanilla meta-learning framework directly produces the mean of prior $\mathbf{w}^P$ by sampling from hyperposterior $\mathcal{Q} = \mathcal{N}(\mathbf{w}^{\mathcal{Q}}, \sigma_{\mathbf{w}}^2 I_{d_{\mathbf{w}}})$. Then the base learner algorithm $A_b(S, P)$ utilizes the sample set $S$ and the prior $P = \mathcal{N}(\mathbf{w}^P, \sigma_{\mathbf{w}}^2 I_{d_{\mathbf{w}}})$ to produce a posterior $Q = A_b(S, P) = \mathcal{N}(\mathbf{w}^Q, \sigma_{\mathbf{w}}^2 I_{d_{\mathbf{w}}})$. Similarly with the two-steps proof in Theorem 3, we first get an intra-task bound by using Theorem 5. For any $\delta_i > 0$, we have

$$
\begin{aligned}
\mathbb{P}_{S_i \sim D_i^{m_i}} & \left\{ \mathop{\mathbb{E}}_{(\mathbf{x}, y) \sim D_i} \mathop{\mathbb{E}}_{\mathbf{w}^P \sim \mathcal{N}(\mathbf{w}^{\mathcal{Q}}, \sigma_{\mathbf{w}}^2 I_{d_{\mathbf{w}}})} \mathop{\mathbb{E}}_{\mathbf{w} \sim \mathcal{N}(\mathbf{w}^Q, \sigma_{\mathbf{w}}^2 I_{d_{\mathbf{w}}})} \ell(h_{\mathbf{w}}(\mathbf{x}), y) \right. \\
\leq & \frac{c_2}{1 - e^{-c_2}} \cdot \frac{1}{m_i} \sum_{j=1}^{m_i} \mathop{\mathbb{E}}_{\mathbf{w}^P \sim \mathcal{N}(\mathbf{w}^{\mathcal{Q}}, \sigma_{\mathbf{w}}^2 I_{d_{\mathbf{w}}})} \mathop{\mathbb{E}}_{\mathbf{w} \sim \mathcal{N}(\mathbf{w}^Q, \sigma_{\mathbf{w}}^2 I_{d_{\mathbf{w}}})} \ell(h_{\mathbf{w}}(\mathbf{x}_j), y_j) \\
& + \frac{1}{(1 - e^{-c_2}) \cdot m_i} \left( \frac{1}{2 \sigma_{\mathbf{w}}^2} \|\mathbf{w}^{\mathcal{Q}}\|^2 + \mathop{\mathbb{E}}_{\mathbf{w}_i^P \sim \mathcal{N}(\mathbf{w}^{\mathcal{Q}}, \sigma_{\mathbf{w}}^2 I_{d_{\mathbf{w}}})} \frac{1}{2 \sigma_{\mathbf{w}}^2} \|\mathbf{w}_i^Q - \mathbf{w}_i^P\|^2 + \log \frac{1}{\delta_i} \right), \forall \mathcal{Q} \left. \right\} \geq 1 - \delta_i,
\end{aligned} \tag{42}
$$

The term $\mathop{\mathbb{E}}_{\mathbf{w}_i^P \sim \mathcal{N}(\mathbf{w}^{\mathcal{Q}}, \sigma_{\mathbf{w}}^2 I_{d_{\mathbf{w}}})} \frac{1}{2 \sigma_{\mathbf{w}}^2} \|\mathbf{w}_i^Q - \mathbf{w}_i^P\|^2$ can be simplified as

$$
\begin{aligned}
& \mathop{\mathbb{E}}_{\mathbf{w}_i^P \sim \mathcal{N}(\mathbf{w}^{\mathcal{Q}}, \sigma_{\mathbf{w}}^2 I_{d_{\mathbf{w}}})} \frac{1}{2 \sigma_{\mathbf{w}}^2} \|\mathbf{w}_i^Q - \mathbf{w}_i^P\|^2 \\
= & \frac{1}{2 \sigma_{\mathbf{w}}^2} \left( \mathop{\mathbb{E}}_{\mathbf{w}^P} \|\mathbf{w}_i^Q\|^2 - 2 (\mathop{\mathbb{E}}_{\mathbf{w}^P} \mathbf{w}_i^Q)^\top \mathbf{w}^{\mathcal{Q}} + \|\mathbf{w}^{\mathcal{Q}}\|^2 + \mathop{\mathbb{V}}_{\mathbf{w}_i^P} [\|\mathbf{w}_i^P\|] \right) \\
= & \frac{1}{2 \sigma_{\mathbf{w}}^2} \left( \| \mathop{\mathbb{E}}_{\mathbf{w}^P} \mathbf{w}_i^Q - \mathbf{w}^{\mathcal{Q}} \|^2 + \sigma_{\mathbf{w}}^2 \right),
\end{aligned} \tag{43}
$$

where $\mathop{\mathbb{V}}_{\mathbf{w}_i^P} [\|\mathbf{w}_i^P\|]$ denotes the variance of $\|\mathbf{w}_i^P\|$. Then we get an inter-task bound. For any $\delta_0 > 0$, we have

$$
\begin{aligned}
\mathbb{P}_{(D_i^{m_i}) \sim \tau, S_i \sim D_i^{m_i}, i=1,\dots,n} & \left\{ \mathop{\mathbb{E}}_{(D, m) \sim \tau} \mathop{\mathbb{E}}_{S \sim D^m} \mathop{\mathbb{E}}_{\mathbf{w}^P \sim \mathcal{N}(\mathbf{w}^{\mathcal{Q}}, \sigma_{\mathbf{w}}^2 I_{d_{\mathbf{w}}})} \mathop{\mathbb{E}}_{\mathbf{w} \sim \mathcal{N}(\mathbf{w}^Q, \sigma_{\mathbf{w}}^2 I_{d_{\mathbf{w}}})} \mathop{\mathbb{E}}_{(x, y) \sim D_i} \ell(h_{\mathbf{w}}(\mathbf{x}), y) \right. \\
\leq & \frac{c_1}{1 - e^{-c_1}} \cdot \frac{1}{n} \sum_{i=1}^{n} \mathop{\mathbb{E}}_{\mathbf{w}^P \sim \mathcal{N}(\mathbf{w}^{\mathcal{Q}}, \sigma_{\mathbf{w}}^2 I_{d_{\mathbf{w}}})} \mathop{\mathbb{E}}_{\mathbf{w} \sim \mathcal{N}(\mathbf{w}^Q, \sigma_{\mathbf{w}}^2 I_{d_{\mathbf{w}}})} \mathop{\mathbb{E}}_{(x, y) \sim D_i} \ell(h_{\mathbf{w}}(\mathbf{x}), y) \\
& + \frac{1}{(1 - e^{-c_1}) n} \left( \frac{1}{2 \sigma_{\mathbf{w}}^2} \|\mathbf{w}^{\mathcal{Q}}\|^2 + \log \frac{1}{\delta_0} \right), \forall \mathcal{Q} \left. \right\} \geq 1 - \delta_0.
\end{aligned} \tag{44}
$$

For any $\delta > 0$, set $\delta_0 \triangleq \frac{\delta}{2}$ and $\delta_i \triangleq \frac{\delta}{2n}$ for $i = 1, \ldots, n$. Using the union bound, we finally get

$$
\mathbb{P}_{(D_i^{m_i}) \sim \tau, S_i \sim D_i^{m_i}, i=1, \ldots, n} \Bigg\{ er(\mathcal{Q})
$$

$$
\leq \frac{c_1 c_2}{(1 - e^{-c_1})(1 - e^{-c_2})} \cdot \frac{1}{n} \sum_{i=1}^{n} \mathop{\mathbb{E}}_{\mathbf{v} \sim \mathcal{N}(\mathbf{v}^{\mathcal{Q}}, \sigma_{\mathbf{v}}^2 I_{d_{\mathbf{v}}}), \mathbf{w}^P = \Phi_{\mathbf{v}}(D), P_i = \mathcal{N}(\mathbf{w}^P, \sigma_{\mathbf{w}}^2 I_{d_{\mathbf{w}}})} \hat{er}(A_b(S_i, P_i))
$$

$$
+ \frac{c_1}{1 - e^{-c_1}} \cdot \frac{1}{n} \sum_{i=1}^{n} \frac{1}{(1 - e^{-c_2}) \cdot m_i} \left( \frac{1}{2\sigma_{\mathbf{w}}^2} \|\mathbf{w}^{\mathcal{Q}}\|^2 + \frac{1}{2\sigma_{\mathbf{w}}^2} \| \mathop{\mathbb{E}}_{\mathbf{w}^P} \mathbf{w}_i^Q - \mathbf{w}^{\mathcal{Q}} \|^2 + \frac{1}{2} + \log \frac{2n}{\delta} \right)
$$

$$
+ \frac{1}{(1 - e^{-c_1})n} \left( \frac{1}{2\sigma_{\mathbf{w}}^2} \|\mathbf{w}^{\mathcal{Q}}\|^2 + \log \frac{2}{\delta} \right), \forall \mathcal{Q} \Bigg\} \geq 1 - \delta. \tag{45}
$$

Similarly, we can further simplify the notation and obtain that

$$
\mathbb{P}_{(D_i^{m_i}) \sim \tau, S_i \sim D_i^{m_i}, i=1, \ldots, n} \Bigg\{ er(\mathcal{Q}) \leq c_1' c_2' \hat{er}(\mathcal{Q})
$$

$$
+ \left( \sum_{i=1}^{n} \frac{c_1' c_2'}{2c_2 n m_i \sigma_{\mathbf{w}}^2} + \frac{c_1'}{2c_1 n \sigma_{\mathbf{w}}^2} \right) \|\mathbf{w}^{\mathcal{Q}}\|^2 + \sum_{i=1}^{n} \frac{c_1' c_2'}{2c_2 n m_i \sigma_{\mathbf{w}}^2} \| \mathop{\mathbb{E}}_{\mathbf{w}^P} \mathbf{w}_i^Q - \mathbf{w}^{\mathcal{Q}} \|^2
$$

$$
+ const(\delta, n, m_i), \forall \mathcal{Q} \Bigg\} \geq 1 - \delta, \tag{46}
$$

where $c_1' = \frac{c_1}{1 - e^{-c_1}}$ and $c_2' = \frac{c_2}{1 - e^{-c_2}}$. This completes the proof.

## C  DETAILS OF EXPERIMENTS

### C.1  DATA PREPARATION

We used the 5-way 50-shot classification setups, where each task instance involves classifying images from 5 different categories sampled randomly from one of the meta-sets. We did not employ any data augmentation or feature averaging during meta-training, or any other data apart from the corresponding training and validation meta-sets.

### C.2  NETWORK ARCHITECHTURE

**Auto-Encoder for LCC** For CIFAR100, the encoder is 7 layers with 16-32-64-64-128-128-256 channels. Each convolutional layer is followed by a LeakyReLU activation and a batch normalization layer. The 1st, 3rd and 5th layer have stride 1 and kernel size $(3, 3)$. The 2nd, 4th and 6th layer have stride 2 and kernel size $(4, 4)$. The 7th layer has stride 1 and kernel size $(4, 4)$. The decoder is the same as encoder except that the layers are in reverse order. The input is resized to $32 \times 32$. For Caltech-256, the encoder is 5 layers with 32-64-128-256-256 channels. Each convolutional layer is followed by a LeakyReLU activation and a batch normalization layer. The first 4 layers have stride 2 and kernel size $(4, 4)$. The last layer has stride 1 and kernel size $(6, 6)$. The decoder is the same as encoder except that the layers are in reverse order. The input is resized to $96 \times 96$.

**Base Model** The network architecture used for the classification task is a small CNN with 4 convolutional layers, each with 32 filters, and a linear output layer, similar to (Finn et al., 2017). Each convolutional layer is followed by a Batch Normalization layer, a Leaky ReLU layer, and a max-pooling layer. For CIFAR100, the input is resized to $32 \times 32$. For Caltech-256, the input is resized to $96 \times 96$.

### C.3  OPTIMIZATION

**Auto-Encoder for LCC** As optimizer we used Adam(Kingma & Ba, 2015) with $\beta_1 = 0.9$ and $\beta_2 = 0.999$. The initial learning rate is $1 \times 10^{-4}$. The number of epochs is 100. The batch size is 512.

**LCC Training** We alternatively train the coefficients and bases of LCC with Adam with $\beta_1 = 0.9$ and $\beta_2 = 0.999$. In specifics, for both datasets, we alternatively update the coefficients for 60 times and then update the bases for 60 times. The number of training epochs is 3.The number of bases is 64. The batch size is 256.

**Pre-Training of Feature Extractor** We use a 64-way classification in CIFAR-100 and 150-way classification in Caltech-256 to pre-train the feature embedding only on the meta-training dataset. For both CIFAR100 and Caltech-256, an L2 regularization term of $5e^{-4}$ was used. We used the Adam optimizer. The initial learning rate is $1 \times 10^{-3}$, $\beta_1$ is 0.9 and $\beta_2$ is 0.999. The number of epochs is 50. The batch size is 512.

**Meta-Training** We use the cross-entropy loss as in (Amit & Meir, 2018). Although this is inconsistent with the bounded loss setting in our theoretical framework, we can still have a guarantee on a variation of the loss which is clipped to $[0, 1]$. In practice, the loss is almost always smaller than one. For CIFAR100 and Caltech-256, the number of epochs of meta-training phase is 12; the number of epochs of meta-testing phase is 40. The batch size is 32 for both datasets. As optimizer we used Adam with $\beta_1 = 0.9$ and $\beta_2 = 0.999$. In the setting with a pre-trained base model, the learning rate is $1 \times 10^{-5}$ for convolutional layers and $5 \times 10^{-4}$ for the linear output layer. In the setting without a pre-trained base model, the learning rate is $1 \times 10^{-3}$ for convolutional layers and $5 \times 10^{-3}$ for the linear output layer. The confidence parameter is chosen to be $\delta = 0.1$. The variance hyper-parameter for prior predictor and base model are $\sigma_{\mathbf{w}} = \sigma_{\mathbf{v}} = 0.01$. The hyperparameter $\alpha_1, \alpha_2$ in LML and ML-A are set to 0.01. For MAML (Finn et al., 2017) and MatchingNet (Vinyals et al., 2016). Both two methods use the Adam optimizer with initial learning rate 0.0001. In the meta-training phase, we randomly split the samples of each class into support set (5 samples) and query set (45 samples). The number of epochs is 100. For MAML, the learning rate of inner update is 0.01.

