# OpenReview forum: "Localized Meta-Learning: A PAC-Bayes Analysis for Meta-Leanring Beyond Global Prior"
_ICLR.cc/2020/Conference — Reject_

### Official Review · AnonReviewer1 · 2019-10-26
**Official Blind Review #1**

**Rating:** 3

**Review:**

Post-rebuttal:
===========
Thank you to the authors for responding to my review, and for adding the comparison to other meta-learning methods besides Amit et al. (2018), which makes it clearer in which settings this technique outperforms purely discriminative approaches (in particular with few tasks & many samples from each task). However, I would assume that not using support-query partitioning for MAML and Matching Nets is likely to reduce their performance.

> "Second, from the algorithm perspective, we use the whole sample set S as the input for LCC-based prior predictor \bar{w}^P = \bar{\Phi}_v(S) ."
Thanks to the authors for this clarification--it is now clear to me that the experimental setup is not the episodic training setup of Vinyals et al. (2016) that partitions episodes into support and query sets, thus the difficulty of comparison to other few-shot learning methods that use this setup.

However, this exposes a potential problem with the formulation: As the authors state in the submission, "...the prior P_i in each task has to be chosen independently of the sample set S_i", in alignment with prior works in data-dependent PAC-Bayes bounds [Catoni, 2007; Parrado-Hernandez et al., 2012; Dziugaite & Roy, 2018]. However, even though the authors begin section 4.1 by stating "Fortunately, the PAC-Bayes theorem allows us to choose prior upon the data distribution D_i. Therefore, we propose a prior predictor...which receives task data distribution Dm and outputs the mean of prior wP"; the prior predictor employed is the "empirical prior predictor" that operates directly on the sample set S_i.

This appears to be a contradiction that is not sufificiently addressed in the text (nor in the response to Reviewer #4). To fix this, the authors would have to more clearly explain why their theoretical results do not require the separation of task-specific data into a subset of data used to produce the prior and a subset used in the computation of the bound, or adapt the experimental setting to meet this theoretical requirement (in which case the setup is very similar to the support-query partitioning commonly used to evaluate few-shot learning methods, therefore bringing into question the necessity of using an alternate evaluation protocol to the one that is standard in few-shot learning).

Before rebuttal:
=============
The submission makes use of a data-dependent PAC-Bayes bound on the generalization error of a classifier estimated in a few-shot learning setup. The episodic few-shot learning setup from Vinyals et al. (2016) provides a small dataset for each task, partitioned into a support and a query set; at test time, only the labels for the support set are provided. The submission takes advantage of this setup by leveraging the support set in the construction of a data-dependent prior, an idea referred to as "locality"; this is in contrast to prior work in PAC-Bayes for hierarchical models (e.g., Pentina & Lampert, 2014; Amit & Meir, 2018) in which the data-dependency enters only across tasks, and not within a task.

Strengths:
- Coherent formulation of data-dependent PAC-Bayes for a meta-learning setting that partitions episodic data into a support set (used to compute the data-dependent prior) and a query set (used to produce the posterior.
- The method outperforms prior approaches constructed from PAC-Bayes generalization bounds (LML; ML-A; ML-AM; ML-PL) on the Caltech-256 and CIFAR-100 datasets.

Weaknesses:
- The framing as "localized meta-learning" obscures the lack of difference from the standard partitioning in few-shot episodes in a support and query set.
- The proposed method makes heavy use of prior machinery (LCC, prototype-based prior predictor), and as such, the algorithmic novelty is limited.
- No comparison is made to approaches that are not constructed use PAC-Bayes generalization bounds (Vinyals et a. 2016; Finn et al. 2017), even though they are readily applied in such settings.

**Experience Assessment:**

I have published one or two papers in this area.

**Review Assessment: Checking Correctness Of Derivations And Theory:**

I assessed the sensibility of the derivations and theory.

**Review Assessment: Checking Correctness Of Experiments:**

I carefully checked the experiments.

**Review Assessment: Thoroughness In Paper Reading:**

I read the paper at least twice and used my best judgement in assessing the paper.

---

> ### Author Response · Authors · 2019-11-15
> **Response to Reviewer #1**
>
> We sincerely appreciate your comments, but we think there is a misunderstanding of our work.  We respond to your main concerns below:
>
> Q: Clarification of our contribution
> A: I would like to clarify that we do not follow the episodic training paradigm from Vinyals et al. to address few-shot learning that splits the meta-training task into support and query. The main contribution of this work is not to propose a specific meta learning algorithm for few shot learning, but to explore the PAC-Bayes meta-learning framework. Compared to the regular meta-learning PAC-Bayes bound in Thm 2, we provide a means to tighten the original PAC-Bayes meta-learning bound by minimizing the task-complexity term. Specifically, we replace the global prior \w^\mathcal{Q} with an LCC-based prior predictor \bar{\Phi}_v(S). Moreover, we propose an LCC-based prior predictor, an implementation of conditional hyperposterior to generate local meta-knowledge for specific task.
>
> Q: lack of difference from the standard partitioning in few-shot episodes in a support and query set.
> A: First, as clarified earlier, we do not adopt the few shot episodic training paradigm. There exist significant differences between few-shot episodes and our environment settings. Specifically, we follow the similar meta-learning environment setting as Amit et al. 2018 to make a fair comparison with PAC-Bayes baselines. That is, meta-learner observes limited number of tasks (from 1 to 11) and each task is not a few-shot problem and we don’t split it into support and query.  In few-shot episode, the meta-training data contains hundreds of thousands tasks and each task is a few-shot learning classification which is split into support and query. Besides, we follow the same joint optimization method as Amit et al. 2018 (to ensure that the benefit of the proposed LML is not from using an improved optimization method but the advantage of localized meta-learning over regular mea-learning). Second, from the algorithm perspective, we use the whole sample set S as the input for LCC-based prior predictor \bar{w}^P = \bar{\Phi}_v(S).
>
> Q:  novelty is limited.
> A: Our main contribution and novelty is the localized meta-learning framework itself and PAC-Bayes analysis.  The proposed LCC-based prior predictor with generalization guarantees is an implementation of that. If we don’t consider theoretical guarantees, as we claimed in Section 3.2, the prior predictor should satisfy three properties: (1) it could learn a mapping function from sample set to base model parameter (2) task-agnostic (3) permutation-invariant. There exist many implementations, such as set transformer [1], relation network [2], task2vec [3]. From the algorithm perspective, in contrast to the existing methods with few-shot episodes, the learned prior \bar{w}^P serves both as an initialization of base model and as a regularizer which restricts the  solution space while allowing variation based on specific task data. It yields a model with smaller error than its unbiased counterpart when applied to a similar task.
>
> [1] Lee,  et al. "Set Transformer: A Framework for Attention-based Permutation-Invariant Neural Networks." 2019.
> [2] Rusu, et al. "Meta-learning with latent embedding optimization." 2018
> [3] Achille et al. "Task2Vec: Task Embedding for Meta-Learning." 2019
>
> Q: Comparison with MAML and MatchingNet
> A: In the experiments, we add two methods, Matching Network  and MAML, which are popular in meta-learning and few-shot learning area for comparison. Our method and other PAC-Bayes baselines outperform these two methods. This is because  MAML and MatchingNet adopt the episodic training paradigm to solve the few-shot learning problem. These two methods requires hundreds of thousands of tasks and each task contains limited samples, which is not the case in our experiment. (To make a fair comparison with PAC-Bayes baselines, we follow the same joint optimization method as Amit et al. 2018, to ensure that the benefit of the proposed LML is not from using an improved optimization method). We also follow the similar meta-learning environment setting as Amit et al. 2018. That is, meta-learner observes limited number of tasks (from 1 to 11) and each task has sufficient samples.) Scarce tasks in meta-training leads to severely meta-overfitting. Moreover, MAML aims to learn a good initialization for base model that can achieve good performance with a few gradient updates. Taking many gradient steps at each task diminishes the effect of the initialization. Therefore, MatchingNet  and MAML is especially suited for few-shot learning with sufficient tasks for meta-training which is not the case in our experiment. In our method, the learned prior serves both as an initialization of base model and as a regularizer which restricts the  solution space while allowing variation based on specific task data. It yields a model with smaller error than its unbiased counterpart when applied to a similar task.

---

### Official Review · AnonReviewer4 · 2019-10-28
**Official Blind Review #4**

**Rating:** 3

**Review:**

In this work, the authors introduce PAC-Bayesian generalization bounds for Meta-Learning. In their framework, they have a hyper-prior distribution, a class of hyper-posteriors and an algorithm A that takes a sample set Si from task Di and a prior P and returns a posterior Q.

Pros:
-- Overall, the main text of the paper is finely written.
-- The motivation is well articulated.
-- The relationship with local coordinate coding seems like an interesting direction.
-- The experiments seem sensible.

The novelty of the bound in Thm. 1:
-- It seems that several aspects of the high-level derivation methodology of the bound are not new. Instead of applying McAllester’s bound twice (as done by Galanti et al. 2016, Amit and Meir 2018), the authors employ Catoni’s bound (a different variant of PAC-Bayes) twice. In addition, the authors apply the standard Gaussian randomization which leads to L2 regularization terms as the capacity terms -- also well known in the literature (see for example Hazan et al. 2013, etc’).
-- I would be happy if the authors point out which parts of their derivations are novel, for instance, the application of local coordinate coding, etc'.

I think the authors' claim that their bound (Thm. 1) is tighter than previously existing bounds is a bit questionable:
-- There are already PAC-Bayesian bounds with the proposed orders of magnitudes with the exact same coefficients by Catoni 2007 and Pascal Germain et al. 2009. In fact, the authors employ Catoni’s bound within the appendix. In my opinion, it should be addressed in the main text.

-- In addition, their bound has two coefficients that are being ignored in their analysis of its magnitude: c/(1-exp(-c)) (here, c stands for c1 or c2) for the error term and 1/(1-exp(-c)) for the capacity term and an additional constant that depends on n,mi and \delta. In the previous bounds, the coefficients are 1 for the error term and 1 for the capacity terms. I think the paper would greatly benefit from a direct comparison between the two.

For instance, as a direct comparison between the two bounds, I would select c, such that, 1/(1-exp(-c)) is close to 1. However, in this case, the coefficient 1/(1-exp(-c)) of the capacity term is huge and makes the overall capacity term very large, which is not favorable. Therefore, it seems to me that the proposed bound is favorable only when the training error is very small (realizable case) and c can be selected to be large. However, when the training error is assumed to be small, it is well known that the gap between the generalization risk and the empirical risk is of order O(capacity/m) (see Thm. 6.8 (bullet 3) in S. Shalev-Schwarz and S. Ben-David 2014). Again, this is also a property of the original bound by Catoni.

Finally, the authors mention that the parameters c1 and c2 control the tradeoff between the empirical error and the capacity terms. I think this is a bit inaccurate, since, c1 and c2 are selected a-priori to the estimation of the error term. Therefore, should be independent of the error term.

-- The presented bound has an additional constant “E_i[const(n,mi,delta)]”.
It is unclear to me what is the magnitude of this quantity. I think it might improve the paper if the authors explicitly analyze this term.
From the proof of Thm. 2 (Eq. 36) it seems to be of order >= ( O_{\alpha,\beta}(\gamma,C) + (1/m_{ik})^{1/2} )^2.
What is the definition of m_{ik} (couldn’t find it, maybe it was defined somewhere -- I think it should have been recalled)? I’m assuming it is mi or something similar.
Second, I’m not sure how to estimate the size of O_{\alpha,\beta}(\gamma,C). From Lem. 1 it depends on some arbitrarily selected quantity \epsilon > 0 and |C| is a function of \epsilon, so I’m not sure how to measure it.
------------------------------------
Soundness:
There are a few things that I'd be happy if clarified regarding Thms. 1 and  2.
-- I’m not sure what is w^Q_i (couldn’t find its definition anywhere). I guess w^Q_i is the center of Q_i = A(Si,P), where P ~ \mathcal{Q}. How can w^Q_i be free within the inequality without being bound to an expectation over P? Especially since you take an expectation over P within the training error. I guess the correct formulation of the bound is one that has  E_{P ~ \mathcal{Q}}[||w^Q_i - w^{\mathcal{Q}}||^2] instead of ||w^Q_i - w^{\mathcal{Q}}||^2.
Maybe the issue is somewhere in between Eq 39 and 40, where the authors take an expectation over Pi (which is not clearly defined as well), but not over Qi?
-- In Eq. 27 last equality: we have a term of the form: E_v ||w^Q - w^P||^2. Where is v in the distance squared?
-- I’m not sure what is the motivation of applying Eq. 28, the bound should be tighter with the LHS term instead of the RHS.
-- In the top half of page 15, P stands for a prior independent of \mathcal{Q}, while in the main text, P~\mathcal{Q}. Therefore, the relationships between different arguments are unclear to me.
-- Finally, I think the paper could be improved if the authors would organize the appendix.


Experiments:
In your experiments you compare the derived algorithm to other PAC-Bayesian baselines. Can you show that your algorithm outperforms other existing baselines in the literature of Meta-Learning (in terms of generalization, etc')?

Typos:
-- “Catino” ---> “Catoni”.

Finally, I think the authors should address the public comment by Tomer Galanti. It seems that Thm. 9 in Galanti et al. (2016) introduces a PAC-Bayesian theorem for Meta-Learning (they call it transfer learning), similar in its nature to Thm. 1 in the current paper. In Thm. 9 they have a hyperprior, denoted by P, learn a hyper posterior Q, for selecting posteriors qi for many different tasks di. In their framework, for each task di, their method returns a posterior qi that minimizes the bound for a specific task. This distribution, qi, is a function of a prior B selected by Q (P in your notation) and the i’th task’s dataset si (Si in your notation). Therefore, instead of denoting it by A(Si,P) as done by the authors, they simply call it a minimizer (but it is a function of the same argument that the authors address). Overall, it seems to me that Galanti et al. had a very similar setting, with different notations and focus on a specific algorithm A.


I think that overall, the paper has interesting insights and the relationship with local coordinate coding seems like an interesting direction. However, I think the paper should not be published in its current form.


**Experience Assessment:**

I have published one or two papers in this area.

**Review Assessment: Checking Correctness Of Derivations And Theory:**

I carefully checked the derivations and theory.

**Review Assessment: Checking Correctness Of Experiments:**

I assessed the sensibility of the experiments.

**Review Assessment: Thoroughness In Paper Reading:**

I read the paper at least twice and used my best judgement in assessing the paper.

---

> ### Author Response · Authors · 2019-11-15
> **Response to Reviewer #4  Part 1**
>
> Thanks for your constructive and valuable comments. We’ve revised our paper following the suggestions and will explain your concerns in the following.
> Q: The novelty of our work, clarification of thm 2 (thm1 in original version) ,  tightness of thm.
> A: Thank you for raising these points and we’d like to elaborate on them. As claimed in line 59-65, our main contribution lies in the PAC-Bayes localized meta-learning framework. Particularly, we propose an LCC-based prior predictor, an implementation of conditional hyperposterior to generate local meta-knowledge for specific tasks. In terms of the novelty of the bound, compared to the regular meta-learning PAC-Bayes bound in Thm 2, we provide a means to tighten the original PAC-Bayes meta-learning bound by minimizing the task-complexity term in Thm 3. Specifically, we replace the global prior \w^\mathcal{Q} with an LCC-based prior predictor \bar{\Phi}_v(S).
>
> We further clarify that the choice of Catoni’s bound over McAllester’s bound itself is not a contribution of this paper. We use Catoni’s bound for fairness. Specifically, we want to make sure that the benefit of the localized meta-learning over regular meta-learning  is not from using a different single-task bound, but from the LCC-based prior predictor. Furthermore, given the advantages of Catoni’s bound (e.g. improved convergence rate, a flexible hyperparameter, etc.), we’d like to first tighten Thm 2 (as compared to Amit et al. 2018 and Pentina et al. 2014), before further comparing it with Thm 3.Thus, we apply Cantoni’s and the universality of Gaussian randomness to derive both regular meta-learning bound in Thm 2 and localized meta-learning bound in Thm 3.
>
> Your comments are very helpful for us to clarify our contribution and choice. We revised the wording in Section 2 and 3 accordingly, and move Catoni’s bound into the main content in Thm 1, to make it more clear and self-contained.
>
> Q: The effect of hyperparameters c_1 and c_2
> A: First, as clarified earlier, the main contribution is not the use of Catoni’s bound but the advantage of localized meta-learning over regular meta-learning. Compared to the regular meta-learning PAC-Bayes bound in Thm2, the main innovation of Thm 3 lies in the task complexity term that  we replace the global prior \w^\mathcal{Q} with an LCC-based prior predictor \bar{\Phi}_v(S)  which exploits the potential to choose an appropriate prior based on task data S to make the task-complexity term small. We present our motivation in line 127-142 and claim that this task-complexity term plays an important role in computing the capacity. Therefore, when we choose the same values of c_1 and c_2 for Thm 2 and Thm 3, we can do a fair comparison and verify the benefits of localized meta-learning. Moreover, similarly with the PAC-Bayes meta-learning framework in Amit et al. 2018, our localized meta-learning framework can also utilize different single-task bound in each of the two steps of proof.
>
> Second, we apologize for missing c^\prime_1 in the coefficients of environment-complexity term and task-complexity term in Thm 2 and Thm 3. We agree with you that setting the coefficients to 1 for both the error term and the capacity term could make a direct comparison between them. Here, we could easily set the coefficients to 1 for both the error term and task-complexity term. Note that the task complexity is the key difference between regular meta-learning bound in Thm 2 and localized meta-learning bound in Thm 3.
>
> Q: “c1 and c2 control the tradeoff ...” is inaccurate.
> A: We agree that c1, c2 are selected a-priori to the estimation of error. However, before observing any training data, we can still set c1 and c2. For instance, we can choose the values of c1 and c2 via validation data. Besides, our expression “parameter c1 control the tradeoff between…” follow the statement below thm 2 in [1].
> [1] Germain, Pascal, et al. A PAC-Bayesian approach for domain adaptation with specialization to linear classifiers 2013.
>
> Q: constant term “E_i[const(n,mi,delta)]” is unclear, analyze about it, definition of m_{ik}
>
> A: Thanks for your suggestion. We move the explicit quantity into the main content in Thm 3 in Eq (12).  This quantity contains two parts, as follows. First, distance between \bar{w}^P (LCC-based prior predictor) defined in Eq. (7) and \hat{w}^P (empirical prior predictor) defined in Eq. (6). We analyzed it below Lemma1 in line173-178. Second, distance between \hat{w}^P(empirical prior predictor) and w^p (expected prior predictor) defined in Eq. (5), we analyzed it below Lemma 2 in line 183-187.
>
> The definition of m_{ik} is below Eq. (5), it means the number of samples for category k in task i.

---

> > ### Author Response · Authors · 2019-11-15
> > **Response to Reviewer #4 Part 2**
> >
> > Q: how to estimate $O_{\alpha,\beta}(\gamma,C)$
> > A: We apologize for missing an inequality w.r.t. |C| in manifold coding theorem in [2] and definition 4 about \episilon-cover. The upperbound for $O_{\alpha,\beta}(\gamma,C)$ and |C| demonstrate that choosing |C| with $O(d_\mathcal{M}\mathcal{N}(\epsilon,\mathcal{M}))$, the approximation error w.r.t.  LCC is bounded by $O(\sqrt{d_\mathcal{M}}\epsilon^2)$. We discuss it in line 174-176.
> >
> > [2] Yu, Kai, Tong Zhang, and Yihong Gong. "Nonlinear learning using local coordinate coding." Advances in neural information processing systems. 2009.
> >
> > Q: Expectation over w^Q_i, definition of w^Q_i,
> > A: We apologize for our mistakes and thanks for pointing them out. W^Q_i is defined in line 116, 149  for regular meta-learning and localized meta-learning, respectively. W^Q_i should be bound to an expectation over P in Thm2 and Thm3.
> >
> > Q: Where is v in E_v ||w^Q - w^P||^2, the motivation of Eq. (31) (Eq. 28 in original version),  P stands for a prior independent of \mathcal{Q} in appendix B.3, while in the main text, P~\mathcal{Q}
> > A: We apologize for the typo in the appendix. We have presented the localized meta-learning in main content with line 143-149 and appendix with line 454-5461.  v denotes the parameter of prior predictor. We first sample v from hyperposterior, then we generate the center of prior of base model w^P through prior predictor w^P=\Phi_v(D).
> >
> > In PAC-Bayes theory, the prior P has to be chosen independently of the sample set but can be dependent on the data distribution D. Therefore, we propose expected prior predictor, that the center of prior distribution w^P is generated by w^P=\Phi_v(D), where v is the parameters for the prior predictor. As we claimed above Eq.(6), the input for expected prior predictor is a data distribution which is considered unknown, and our only insight as to D is through sample set S. So we approximate \Phi_v(D) with LCC-based prior predictor \Phi_v. Thanks for pointing it out. We add a paragraph above Eq. (31) in appendix to make it more clear.
> >
> > We claim that P~\mathcal{Q}, but we do not make any constraint about the independency of P in line442-461. The prior is a reference and the bound still holds when prior depends on posterior in PAC-Bayes theory. More similar setting can be found in localized pac-bayes learning as we introduced in Sec5, i.e. [3,4]. Moreover, the proof of Amit and Meir 2018 of PAC-Bayes meta-learning bound  in appendix A.1 also follows this setting.
> > [3] Dziugaite et al. Data-dependent PAC-Bayes priors via differential privacy. 2018.
> > [4] Lever et al. Tighter pac-bayes bounds through distribution-dependent priors." 2013
> >
> > Q: reorganize the appendix.
> > A: Thanks for your suggestion.  We reorganized the appendix, add an abstract at the beginning of the appendix.
> >
> > Q: compare with other meta-learning baselines.
> > A: Thanks for your suggestion. In our experiments, we added two methods, Matching Network [5] and MAML [6], which are popular in meta-learning and few-shot learning area for comparison. Our method and other PAC-Bayes baselines outperform these two methods. This is because  MAML and MatchingNet adopt the episodic training paradigm to solve the few-shot learning problem. These two methods requires hundreds of thousands of tasks and each task contains limited samples, which is not the case in our experiment. (To make fair comparison with PAC-Bayes baselines, we follow the same joint optimization method as Amit et al. 2018 (guarantee that the benefit of the proposed LML is not from using an improved optimization method). We also follow the similar meta-learning environment setting as Amit et al. 2018. That is, meta-learner observes limited number of tasks (from 1 to 11) and each task has sufficient samples.) Scarce tasks in meta-training leads to severely meta-overfitting. Moreover, MAML aims to learn a good initialization for base model that can achieve good performance with a few gradient updates. Taking many gradient steps at each task diminishes the effect of the initialization. Therefore, MatchingNet  and MAML is especially suited for few-shot learning with sufficient tasks for meta-training which is not the case in our experiment. In our method, the learned prior serves both as an initialization of base model and as a regularizer which restricts the  solution space while allowing variation based on specific task data. It yields a model with smaller error than its unbiased counterpart when applied to a similar task.
> >
> > Q: typo, the work of Galanti et al. (2016)
> > A: Thanks for pointing it out. We added it in our related work. It would be interesting for considering non i.i.d setting for localized meta-learning from the transfer learning perspective.

---

> > > ### Comment · AnonReviewer4 · 2019-11-15
> > > **Revision**
> > >
> > > Could you please upload the revised version, so I can give it a second look?
> > >
> > > Thanks.

---

> > > > ### Author Response · Authors · 2019-11-15
> > > > **Revision**
> > > >
> > > > Thank you for helping us improve the paper! We have uploaded it.

---

### Official Review · AnonReviewer2 · 2019-10-31
**Official Blind Review #2**

**Rating:** 8

**Review:**

This paper discusses a theoretical analysis of localized meta-learning. The authors are motivated from a PAC-Bayes perspective of meta-learning wherein the system learns a hyper-prior on the hypothesis space using the posteriors of the weak-learners learnt on their individual tasks. The first contribution of the paper is to construct a PAC-Bayes generalization bound for the case when the hyper-prior and the prior on the new task are both isotropic Gaussian. The second contribution is to develop a localized meta-learning algorithm which predicts a better prior for the weak-learner using the data distribution of the new task. The authors use ideas from metric learning to learn the predictor for the prior using Local Coordinate Coding.

I would like to accept this paper. It has very clear contributions to meta-learning. I expect the core idea of splitting a single hyper-prior into anchor-points to have good impact on real problems because the task diversity in most realistic meta-learning scenarios makes it difficult to learn a meaningful hyper-prior.

I have a few minor comments which I would like the authors to address.

1. Choosing the prior w^P using data from the new task will work well if one has access to a few support samples. How do your experiments reflect this? I think the numerical results in Figure 3-4 are high because the shot is abnormally high (it is 50). It is possible to simply fine-tine a pre-trained network to near perfect accuracy with this much data. This is a major drawback of the predictor for the prior.
2. Can you show results for more standard settings, e.g., 5-way, 5-shot? Can you show results on mini-ImageNet?
3. One way to build upon Theorem 2 would be to use the generalization error of the predictor Phi_{v^Q}(S) and characterize the number of support samples one necessary for a given number of anchor points.
4. The authors note in Section 5.1 that the meta-learning objective is difficult to optimize. It is unsatisfying to use the pre-trained model as an initialization for the meta-training. Can you elaborate more on this?

**Experience Assessment:**

I have published one or two papers in this area.

**Review Assessment: Checking Correctness Of Derivations And Theory:**

I assessed the sensibility of the derivations and theory.

**Review Assessment: Checking Correctness Of Experiments:**

I carefully checked the experiments.

**Review Assessment: Thoroughness In Paper Reading:**

I read the paper thoroughly.

---

> ### Author Response · Authors · 2019-11-15
> **Response to Reviewer #2 part1**
>
> We deeply appreciate the reviewer for the positive remarks, constructive suggestions and the interest. We’ve revised our paper following the suggestions and will explain your concerns in the following.
>
> Q: handle tasks with limited samples? results for standard setting, 5-way5-shot
> A: First, we would like to clarify that, to make fair comparison with PAC-Bayes baselines, we follow the same joint optimization method as Amit et al. 2018 (to ensure  that the benefit of the proposed LML is not from using an improved optimization method). This joint optimization method is not meant for problems with hundreds of thousands of training tasks, as typically required in few-shot learning. Thus, we follow the similar meta-learning environment setting as Amit et al. 2018. That is, meta-learner observes limited number of tasks (from 1 to 11) and each task contains sufficient samples (eg. 50 per class).
>
> Nevertheless, to strengthen our experiments, we added two more methods, Matching Network  and MAML, which are popular in meta-learning and few-shot learning area for comparison. Our method and other PAC-Bayes baselines outperform these two methods. Note that  MAML and MatchingNet adopts the episodic training paradigm to solve the few-shot learning problem. The meta-training process requires hundreds of thousands of tasks and each task contains limited samples, which is not the case in our experiment.  Scarce tasks in meta-training leads to severely meta-overfitting. Therefore, MatchingNet  and MAML is especially suited for few-shot learning with sufficient tasks for meta-training. In our method, the learned prior serves both as an initialization of base model and as a regularizer which restricts the  solution space while allowing variation based on specific task data. It yields a model with smaller error than its unbiased counterpart when applied to a similar task.  We further tested the few-shot few-task setting (5 shot 11 task).  The results are as follows
> Caltech				                       Cifar
> without pretrain	with pretrian	       without pretrain	with pretrian
> LML	63.7		69.0		               61.5			65.8
> ML-A	59.3		65.5		               58.9			62.9
> MAML	48.1		52.2		                47.6			53.9
>  Although the performance of all methods drops,  our method still outperforms all other baselines.
>
> Second, we agree that it is an interesting problem to apply localized meta-learning for few-shot learning problem, like miniImagenet. As shown in Thm 3 and Thm 2, the derived generalization error for both localized meta-learning and regular meta-learning converge at the rate of O(1/nm), n is the number of tasks, m is the samples per task. We can use Thm 2 to explain why few-shot episodic meta-learning methods, like MAML, work.  It indicates that even if each task contains very few samples, the generalization error is small if the meta-training set contains a large number of tasks. Besides, the proposed LCC-based prior predictor essentially approximate the expected prior predictor with a compact set of anchor points, which has a linear complexity w.r.t. data size. Therefore, we think localized meta-learning and the prior predictor is indeed a promising solution for few-shot learning.
> The bottleneck of tackling few-shot learning problem is not the proposed prior predictor but the scalability of joint optimization methods in Amit et al. 2018. The generalization bound of localized meta-learning is quite general to derive different optimization method. One potential solution is to adopt the amortized variational inference method like [1] to address the scalability issue for training hundreds of thousands of tasks. We intent to investigate it as a future work.
> [1]  Ravi et al. "Amortized bayesian meta-learning." (2018).
>
> Q: use generalization error of prior predictor to characterize samples per task, number of anchor points.
> A: We agree that Thm 3 (Thm 2 in original version) shows the relationship between generalization error and the number of tasks n, samples per task m in the task-complexity term $\|E_v w^Q- \bar{\Phi}_v(S)\|$ for localized meta-learning and $\|Ew^Q-W^\mathcal{Q}\|$ for regular meta-learning.
> As shown in Thm 3,  the derived generalization error converges at the rate of (1/nm).  It indicates that even if each task contains very few samples, the generalization error is small if the meta-training set contains sufficient number of tasks.
> The number of anchor points is related to the LCC-based prior predictor. Lemma 1 demonstrates that the approximation error of LCC depends on the intrinsic dimension of the manifold instead of the dimension of input. As [2] claimed below thm 2.1, “LCC does not require the data to lie precisely on a manifold, and it does not require knowing the intrinsic dimension of the manifold. In fact, similar results hold even when the data only approximately lie on a manifold.” In practice, a small |C| is often sufficient.
> [2] Yu, Kai et al. Nonlinear learning using local coordinate coding 2009.

---

> > ### Author Response · Authors · 2019-11-15
> > **Response to Reviewer #2 part 2**
> >
> > Q: unsatisfying to use pretrain model as initialization.
> > A: Thanks for pointing it out. We revise the related wording. We aim to design different meta-learning environment settings (i.e., with or without pre-trained model) to verify the efficacy of localized meta-learning. In Figure 3, it shows that LML consistently outperforms other PAC-Bayes baselines in both settings.
> >
> > In the meta learning framework, meta model extracts meta-knowledge as a prior to improve the learning of base model for new task. We consider the pre-trained model as a data-dependent hyperprior for meta-training. In our framework, if the distance between the hyperprior and hyperposterior (the learned meta model) is small, it will improve the generalization performance (reduce the environment complexity). This has been verified in our experiment that the methods with pre-train model consistently outperform those without pre-train model. We added this explanation in Section 5.

---

### Official Review · AnonReviewer3 · 2019-10-31
**Official Blind Review #3**

**Rating:** 6

**Review:**

### Summary
​
This paper proposes a tight bound in generalization to new tasks in meta-learning framework, by controlling the task prior with Local Coordinate Coding (LCC) prediction. In classification tasks, the algorithm using this bound demonstrates superior performance over other meta-learning methods which are based on PAC-Bayes bounds, but without the proposed prior prediction.
​
​
### Strengths
- The paper is well written, and maintains a logical flow with the proofs and inference from them.
- The idea and intuition for using a learned prior is sound, and is backed by PAC-Bayes theory.
- Proposing a tighter generalization bound O(1/m) as opposed to existing bounds of O(1/sqrt(m)) is a meaningful contribution and its efficacy is well shown in the results.
​
### Weaknesses
- Could the authors comment on how their LCC-basedd prior prediction can be extended to other meta learning setups like regression and reinforcement learning?
- The baselines compared with are other PAC-Bayes bounds and successfully justifies the contribution. Could the authors provide a comparison with other meta-learning methods (like [1]) to have a holistic view of where this proposed bound gets this line of work?
​
​
#### Minor:
- Spellings: "pratical" -> "practical" (pg1, abstract); "varible" -> "variable" (pg 3); "simplifies" -> "simplify" (pg6, optimization of LLC)
- [2] seems to be a related work, as instead of using the global prior, they identify the task first (similar to localized prior), and then utilize it for better performance.
​
### References
[1] Finn, Chelsea, Pieter Abbeel, and Sergey Levine. "Model-agnostic meta-learning for fast adaptation of deep networks." Proceedings of the 34th International Conference on Machine Learning-Volume 70. JMLR. org, 2017.
​
[2] Vuorio, R., Sun, S. H., Hu, H., & Lim, J. J. (2018). Toward multimodal model-agnostic meta-learning. arXiv preprint arXiv:1812.07172.
​
​
### Score
6 - Weak Accept

**Experience Assessment:**

I have read many papers in this area.

**Review Assessment: Checking Correctness Of Derivations And Theory:**

I assessed the sensibility of the derivations and theory.

**Review Assessment: Checking Correctness Of Experiments:**

I assessed the sensibility of the experiments.

**Review Assessment: Thoroughness In Paper Reading:**

I read the paper at least twice and used my best judgement in assessing the paper.

---

> ### Author Response · Authors · 2019-11-15
> **Response to Reviewer #3**
>
> We greatly appreciate your comments and thoughtful suggestions.  We respond to each comment as follows.
>
> Q: extend to regression and reinforcement learning problem
> A: Thank you for your suggestion. In our work, the PAC-Bayes analysis is only formulated for the classification problem. It would be interesting to extend it to regression and reinforcement learning. We add this discussion in our future work. Indeed, the high level idea of localized meta-learning, that meta-model provides a task data-dependent prior for the learning of new tasks, can be applied to regression and reinforcement learning problems. As we claimed in Section 3.2, the prior predictor should satisfy three properties: (1) it could learn a mapping function from sample set to base model parameter (2) task-agnostic (3) permutation-invariant.  Take regression problem as an example, we can still use LCC as the mapping function between sample set and model parameter. However, the nearest mean classifier only works in classification problem. Instead, we can replace it with set transformer [1], relation network [2] or task2vec [3] as a substitute.
>  [1] Lee, Juho, et al. Set Transformer: A Framework for Attention-based Permutation-Invariant Neural Networks. 2019.
> [2] Rusu, Andrei A., et al. Meta-learning with latent embedding optimization. 2018
> [3] Achille, Alessandro, et al. Task2Vec: Task Embedding for Meta-Learning. 2019
>
> Q: comparison with other meta-learning methods like MAML.
> A: A: In experiments, we added two methods, Matching Network and MAML, which are popular in meta-learning and few-shot learning area for comparison. Our method and other PAC-Bayes baselines outperform these two methods. This is because  MAML and MatchingNet adopt the episodic training paradigm to solve the few-shot learning problem. These two methods require hundreds of thousands of tasks and each task contains limited samples, which is not the case in our experiment. (To make a fair comparison with PAC-Bayes baselines, we follow the same joint optimization method as Amit et al. 2018 to ensure that the benefit of the proposed LML is not from using an improved optimization method. Thus, we also follow the similar meta-learning environment setting as Amit et al. 2018. That is, meta-learner observes limited number of tasks (from 1 to 11) and each task has sufficient samples.) Scarce tasks in meta-training leads to severely meta-overfitting. Moreover, MAML aims to learn a good initialization for base model that can achieve good performance with a few gradient updates. Taking many gradient steps at each task diminishes the effect of the initialization. Therefore, MatchingNet  and MAML is especially suited for few-shot learning with sufficient tasks for meta-training which is not the case in our experiment. In our method, the learned prior serves both as an initialization of base model and as a regularizer which restricts the  solution space while allowing variation based on specific task data. It yields a model with smaller error than its unbiased counterpart when applied to a similar task.
>
> Q: typos and related work
> A: Thanks for pointing out the spelling mistakes and the related work. [2] proposes a multimodal MAML to handle diverse task distribution for few-shot problem. Our work is motivated by the PAC-Bayes meta-learning framework. We revised our wording and added the suggested reference in related work.

---

### Public Comment · ~Tomer_Galanti1 · 2019-10-03
**A Theoretical Framework for Deep Transfer Learning**

We like your work. Please consider citing "A Theoretical Framework for Deep Transfer Learning" by Galanti et al. 2016, which introduces generalization bounds for transfer learning and PAC-Bayesian bounds in particular. This does not impact the overall novelty of your work.

---

> ### Author Response · Authors · 2019-11-15
> **Thanks for your interest**
>
> A: Thanks for pointing it out. We added it in our related work. It would be interesting for considering non i.i.d setting for localized meta-learning from the transfer learning perspective.

---

### Decision · Program_Chairs · 2019-12-19

**Decision:**

Reject

**Comment:**

This paper proposes PAC-Bayes bounds for meta-learning. The reviewers who are most knowledgeable about the subject and who read the paper most closely brought up several concerns regarding novelty (especially a description of how the proposed bounds relate to those in prior works (Pentina el al. (2014), Galanti et al. (2016) and Amit and Meir (2018))) and regarding clarity. The reviewers found theoretical analysis and proofs hard to follow. For these reasons, the paper isn't ready for publication at this time. See the reviewer's comments for details.